



# Coupling physics and biogeochemistry thanks to high resolution observations of the phytoplankton community structure in the North-Western Mediterranean Sea

Pierre Marrec[1], Andrea M. Doglioli[1], Gérald Grégori[1], Mathilde Dugenne[1], Alice Della Penna[1], Nagib Bhairy[1], Thierry Cariou[2], Sandra Hélias Nunige[1], Soumaya Lahbib[1], Gilles Rougier[1], Thibaut Wagener[1] and Melilotus Thyssen[1]

[1]Aix Marseille Univ, Université de Toulon, CNRS, IRD, MIO UM 110, Marseille, France, 13288, Marseille, France.
[2]Sorbonne Universités, UPMC Univ Paris 06, CNRS, Fédération de Recherche (FR2424), Station Biologique de Roscoff, 29680, Roscoff, France.

*Correspondence to*: Pierre Marrec (pierre.marrec@mio.osuphyteas.fr)

**Abstract.** Fine-scale physical structures and ocean dynamics strongly influence and regulate biogeochemical and ecological processes. These processes are particularly challenging to describe and understand because of their ephemeral nature. The OSCAHR (Observing Submesoscale Coupling At High Resolution) campaign has been conducted in fall 2015 in which, a fine-scale structure in the North Western Mediterranean Ligurian subbasin was pre-identified using both satellite and numerical modeling data. Along the ship track, various variables were measured at the surface (temperature, salinity, chlorophyll-*a* and nutrients concentrations) with ADCP current velocity. We also deployed a new model of CytoSense automated flow cytometer (AFCM) optimized for small and dim cells, for near real-time characterization of surface phytoplankton community structure of surface waters with a spatial resolution of few km and a hourly temporal resolution. For the first time with this type of AFCM we were able to resolve *Prochlorococcus* and *Synechococcus* picocyanobacteria. The vertical physical dynamics and biogeochemical properties of the studied area were investigated by continuous high resolution CTD profiles thanks to a moving vessel profiler (MVP) during the vessel underway associated to a 1-m vertical resolution pumping system deployed during fixed stations. The observed fine-scale feature presented a cyclonic structure with a relatively cold core surrounded by warmer waters. Surface waters were totally depleted in nitrate and phosphate. In addition to the doming of the isopycnals by the cyclonic circulation, an intense wind event induced Ekman pumping. The upwelled subsurface cold nutrient-rich water fertilized surface waters, characterized by an increase in Chl-*a* concentration. *Prochlorococcus*, pico- and nano-eukaryotes were more abundant in cold core waters while *Synechococcus* dominated in warm boundary waters. Nanoeukaryote were the main contributors



(>50%) in terms of pigment content (FLR) and biomass. Biological observations based on the mean cell's red fluorescence recorded by AFCM combined with physical properties of surface waters suggest a distinct origin for two warm boundary waters. Finally, the application of a matrix growth population model based on high-frequency AFCM measurements in warm boundary surface waters provides estimates of in-situ growth rate and

apparent net primary production for *Prochlorococcus* ($\mu$=0.21 d$^{-1}$, NPP=0.11 mgC.m$^{-3}$.d$^{-1}$) and *Synechococcus* ($\mu$=0.72 d$^{-1}$, NPP=2.68 mgC.m$^{-3}$.d$^{-1}$), which corroborate their opposite surface distribution pattern. The innovative adaptive strategy applied during OSCAHR with a combination of several multidisciplinary and complementary approaches involving high-resolution *in-situ* observations and sampling, remote-sensing and model simulations provided a deeper understanding of the marine biogeochemical dynamics through the first

trophic levels.

## 1 Introduction

Despite representing only 0.2% of the global photosynthetically active carbon (C) biomass, phytoplankton accounts for about the half of global primary productivity on Earth (Falkowski et al. 1998; Field et al., 1998). It forms the base of the marine food web and exerts a major control on global biogeochemical

cycles. In a context of global change, mainly due to the raise of anthropogenic atmospheric $CO_2$ (IPCC, 2013), marine phytoplankton plays a fundamental role in the global C cycle by photosynthetically fixing $CO_2$ and exporting it into the ocean's interior by the biological pump (De La Rocha and Passow, 2007). Phytoplankton community structures are highly heterogeneous over the ocean in terms of assemblage, physiology and taxonomy (Barton et al., 2010; De Vargas et al., 2015). Phytoplankton cell volume spans on more than nine

orders of magnitude (Maranon et al., 2015), from *Prochlorococcus* cyanobacteria (~10$^{-1}$ $\mu$m$^3$) to the largest diatoms (>10$^8$ $\mu$m$^3$). Phytoplankton diversity is primarily controlled by environmental factors as, i.e. temperature, nutrients, light availability, vertical stability, predation, which lead to a biogeography of phytoplankton diversity landscape (Lévy et al., 2015). While at a basin scale the phytoplankton community structure is relatively well constrained, at a fine-scale both modeling (Lévy et al., 2001; Clayton et al., 2013;

Lévy et al., 2014; d'Ovidio et al., 2015) and observation (Claustre et al., 1994; d'Ovidio et al., 2010, Clayton et al., 2014; Martin et al., 2015, Cotti-Rausch et al., 2016) have revealed during the last decades that phytoplankton community structure exhibits strong fine-scale variability, i.e. 1-100km (Levy et al., 2015).

Mesoscale (10-100 km, 10-100 days) and submesoscale (1-10 km, 1-10 days) physical dynamics strongly influence and regulate biogeochemical and ecological processes (McGuillicuddy et al., 1998; Levy and



Martin, 2013; McGuilicuddy, 2016). This can have a significant impact on primary productivity (Oschlies and Garçon, 1998; Mahadevan, 2016) and thus on the biological C pump (Levy et al., 2013) and associated export (Siegel et al., 2016). Mesoscale eddies modify the vertical structure of the water column: cyclones and anti-cyclones respectively shoal and deepen isopycnals (McGuillicuddy et al., 1998). Eddy pumping may have a

significant biogeochemical impact in oligotrophic areas (Falkowski et al., 1991): shoaling isopycnals in the center of a mesoscale cyclonic eddy can stimulate phytoplankton productivity by lifting nutrients into the euphotic zone. Eddy stirring and trapping further influence biogeochemical and ecological processes (McGuillicuddy, 2016 for a review). Submesoscale dynamics enhance the supply of nutrients in the euphotic zone in nutrient depleted areas and also influence the light exposure of phytoplankton by modifying density

gradient in the surface layer, which contribute significantly to phytoplankton production (Mahadevan, 2016) and community structure variability (Cotti-Rausch et al., 2016). The underlying biogeochemical submesoscale processes are particularly challenging to describe and understand because of their ephemeral nature. For the moment, submesoscale dynamics have been predominately investigated through the analysis of numerical simulation. The lack of *in-situ* observations at an appropriate spatio-temporal resolution makes difficult the

integration of these *in-situ* data with the model simulations, and it remains still unclear how these processes affect the global state of the ocean (Mahadevan, 2016).

       The Mediterranean Sea represents only ~0.8% in surface and ~0.3% in volume as compared to the World Ocean, but hosts between 4% and 18% of world marine species, making it a biodiversity hotspot (Bianchi and Morri, 2000, Lejeusne et al, 2009). The Mediterranean Sea is a reduced-scale laboratory basin for the

investigation of processes of global importance (Malanotte-Rizzoli et al., 2014; Pascual et al., 2017) because it is characterized by a complex circulation scheme including deep water formation and intense mesoscale and submesoscale variability (Millot and Taupier-Letage, 2005). Mesoscale and submesoscale variability overlays and interacts with the basin and sub-basin scales, producing intricate processes representative of a complex and still unresolved oceanic systems (Malanotte-Rizzoli et al., 2014; Pascual et al., 2017). The small size of the

Mediterranean Sea and the proximity of numerous marine observatories are other outstanding advantages giving its status of 'miniature ocean' laboratory. The Mediterranean Sea is considered as an oligotrophic basin (Moutin and Prieur, 2012) and its primary production by phytoplankton is generally low (D'Ortenzio and Ribera d'Alcala, 2009).

       General surface circulation pattern in the Western basin of the Mediterranean Sea is characterized by

Modified Atlantic Waters (MAW) transported from the Algerian basin to the Ligurian subbasin (Millot and Taupier-Letage, 2005), flowing in the surface and northward from the West part of Corsica called the Western



Corsican Current; and joining the Eastern Corsican Current at the vicinity of the Cap Corse to form the Northern Current (Astraldi and Gasparini, 1992; Millot, 1999). A cyclonic gyre is generated by a recirculation of the Northern Current towards the Western Corsican Current. Our study area was located in the centre of a cyclonic recirculation within the Ligurian subbasin and forced by atmospheric-climatic conditions (Astraldi et al., 1994).

The Ligurian subbasin hydrological regime varies from intense winter mixing to strong thermal stratification in summer and fall. The phytoplankton biomass increases significantly in late winter/early spring, sustained by nutrient fertilization from deep waters, and decreases along with biological activity in summer and fall due to nutrient (N and P) depleted surface waters (Marty et al., 2002). In late summer/early fall season (the time of this present study) the phytoplankton community structure in the Ligurian subbasin is dominated by small size

phytoplankton species (such as *Prochlorococcus*, *Synechococcus*, pico- and nano-eukaryotes; Marty et al., 2008).

The efficient study of submesoscale structures and their associated physical-biological-biogeochemical mechanisms requires the use of a combination of several complementary approaches involving *in-situ* observations and sampling, remote-sensing and model simulations (Pascual et al., 2017). High-resolution

measurements are mandatory to assess the mechanisms controlling the fine-scale biophysical interactions. They are now available thanks to the recent progress in biogeochemical sensor developments, the combination of ship-based measurements and autonomous platforms, and innovative adaptive approaches. The OSCAHR campaign (Observing Submesoscale Coupling At High Resolution, Doglioli and Grégori, 2015, 29/10/2015 – 06/11/2015) aimed at identifying and characterizing such a fine-scale dynamical structure in the North Western

Mediterranean Sea and at studying its influence on the biogeochemical processes, phytoplankton community structure and dynamics at high resolution. In the present study the terms "high resolution" and "fine-scale" aim to describe observations and mechanisms, respectively, and are preferred to the "submesoscale" term.

During the OSCAHR cruise novel platforms for coupling physical-biological-biogeochemical observations and sampling the ocean surface layer at a high spatial and temporal frequency were coupled with

real time analyses of satellite ocean color imagery and altimetry. This real time was necessary for the adaptive strategy mandatory to define and follow *in-situ* the fine-scale dynamical structure identified before the cruise and characterize its evolution along the cruise and after thanks to satellite and numerical modeling data combined to *in-situ* measurements. In this article, we first describe the hydrological structure and dynamics of the studied feature based on satellite data and continuous sea surface measurements. Then we address the corresponding

phytoplankton community structure and distribution based on analyses performed at the single cell level and at high spatio-temporal resolution on an autonomous way. Moreover, we also present the fine-scale vertical



variability of the phytoplankton community structure in various stations within and outside the studied structure resulting in a three dimensional dataset for the investigation of the physical driving mechanisms acting on the phytoplankton community structure. Finally, thanks to the outstanding potential of single cell analysis performed by automated high-resolution flow cytometry we estimate *in-situ* growth rates and address the apparent primary

productivity of the two dominant phytoplankton species (in terms of abundances), *Prochlorococcus* and *Synechococcus* in relation with their environment.

## 2 Materials and Methods

### 2.1 OSCAHR outlines

The OSCAHR research voyage was carried out between the 29/10/2015 and the 06/11/2015 in the

western Ligurian subbasin onboard the R/V Téthys II. A first leg sampled the coastal waters, a second one was dedicated to offshore waters in >1000 m water column area. The present study focuses on the second leg held from the 3$^{rd}$ of November to the 6$^{th}$ of November (Fig. 1). The cruise strategy used an adaptive approach based on the near-real time analysis of both satellite and numerical modeling data to identify dynamical features of interest and follow their evolution. This task was performed thanks to the software package SPASSO (Software

Package for an Adaptive Satellite-based Sampling for Ocean campaigns, http://www.mio.univ-amu.fr/SPASSO) that performs Eulerian and Lagrangian diagnostics of the altimetry-derived currents (d'Ovidio et al, 2015) together with sea surface temperature and chlorophyll-*a* (Chl-*a*) concentration. We sampled a fine-scale dynamical structure characterized by a patch of cold surface water surrounded by warm waters. We recorded physical, biological and chemical data at high frequency (minute to hourly scale) with a combination of

classical (thermosalinograph (TSG), discrete surface sampling) and innovative (automated high-frequency flow-cytometry (AFCM), Moving Vessel Profiler (MVP)) methods. Regular fixed station measurements (classical conductivity, temperature, depth (CTD) profiles and sampling at high vertical resolution (at a meter scale)) were also performed at strategic sampling sites.

### 2.2 Satellite and model products

We used the altimetry-derived (i.e. geostrophic) velocities distributed by AVISO as multi-satellite Mediterranean regional product (http://www.aviso.altimetry.fr) on a daily basis with a spatial resolution of ¼°. Sea surface temperature (SST, level 3 and 4, 1 km resolution) and Chl-*a* concentrations (level 3, 1 km resolution, MODIS-Aqua and NPP-VIIRS sensors) were provided by CMEMS (Copernicus Marine Environment





Monitoring Service, htpp://marine.copernicus.eu). Chl-*a* product is optimized to work in "case 1 waters" (Morel et al., 2006), i.e. open ocean conditions where the optical signal is dominated by phytoplankton. The atmospheric numerical model WRF (Weather Research and Forecasting, Skamarock et al., 2008) provided meteorological forecast (wind speed and direction, irradiance). WRF has been implemented at the Observatory of Universe

Sciences Institut Pytheas (Marseille) as an operational model. Ekman pumping was calculated from the curl of the wind stress: $w = curl(\tau/\rho.f)$, where $w$ is an estimate of the vertical velocity ($w>0$ refers to vertical velocity), $\rho$ is the density of the water, here considered $\rho = 1028$ kg.m$^3$ and $f$ is the Coriolis parameter that is variable with latitude and in the region of study is ~10-4 rad.s$^{-1}$.

### 2.3 Nutrients and Chl-*a* analysis

Nutrient samples were collected in 20 cm$^3$ high-density polyethylene bottles poisoned with HgCl$_2$ to a final concentration of 20 mg.dm$^{-3}$ and stored at 4°C before being analysis in the laboratory a few months later. Nutrient concentrations were determined using a Seal AA3 auto-analyzer following the method of Aminot and Kérouel (2007) with analytical precision of 0.01 µmol.dm$^{-3}$ and quantification limits of 0.02, 0.05 and 0.30 µmol.dm$^{-3}$ for phosphate, nitrate (and nitrite) and silicate, respectively.

To determine Chl-*a* concentrations, $500 \pm 20$ cm$^3$ of seawater were filtered through 47 mm glass-fiber filters (Whatman$^{®}$ GF/F) and immediately frozen at -20°C. Samples were extracted in 5 cm$^3$ of acetone. Correction for phaeopigments was carried out using the acidification method with an HCl solution, after primary fluorescence measurements using a fluorometer (Turner Designs model 10 AU digital fluorometer) to calculate Chl-*a* concentrations according to EPA (1997).

### 20  2.4 Bench top flow cytometry

Seawater samples collected from the Niskin bottles were pre-filtered through a 100 µm mesh size net to prevent any clogging of the flow cytometer. Cryovials (5 cm$^3$) were filled with subsamples that were preserved with glutaraldehyde 0.2 % final concentration for ultraphytoplankton analysis. Samples were then rapidly frozen in and stored in liquid nitrogen until analysis at the PRECYM flow cytometry platform of the institute. In the

laboratory, cryovials were rapidly thawed at room temperature and analyzed using the FACSCalibur flow cytometer (BD Biosciences$^{®}$) of PRECYM. This flow cytometer is equipped with a blue (488 nm) air-cooled argon laser and a red (634 nm) diode laser. For each particle analyzed (cell), five optical parameters were recorded: forward and right angle light scatter, and green (515–545 nm), orange (564–606 nm) and red (653–669 nm) fluorescence wavelength ranges. Data were collected using the CellQuest software (BD Biosciences$^{®}$). The



analysis and identification of ultraphytoplankton groups were performed a posteriori using SUMMIT v4.3 software (Beckman Coulter). For each sample the runtime of the flow cytometer was set up at 5 min. The sample flow rate was about 100 mm$^{-3}$.min$^{-1}$ (corresponding to the "Hi" flow rate of the flow cytometer).

Various ultraphytoplankton groups were optically resolved without any staining on the basis of their

light scatter and fluorescence properties (defined below in the Results Section). Separation of picoeukaryotes and nanophytoplankton was performed by adding 2 µm yellow-green fluorescent cytometry microspheres (Fluoresbrite YG 2 µm, Polyscience Inc.) to the samples. Trucount™ calibration beads (Becton Dickinson Biosciences) were also added to the samples as an internal standard both to monitor the instrument stability and to determine the volume analyzed by the instrument. This is mandatory to compute the cell abundances.

**2.5 Underway surface measurements**

The *in-situ* velocity of the currents was measured by a hull-mounted RDI Ocean Sentinel 75 kHz ADCP (Acoustic Doppler Current-meter Profiler). The configuration used during the whole cruise was: 60 cells, 8 m depth beams, and 1 min averaged. The depth range extended from 18.5 m to 562.5 m.

The onboard surface-water flow-through system pumped seawater at 2 m depth with a flow rate

carefully maintained at 60 dm$^3$.min$^{-1}$. The TSG, a SeaBird SBE21, acquired sea surface temperature (SST) and salinity (SSS) data every minutes during the whole cruise. A Turner Designs fluorometer (10-AU-005-CE) recorded simultaneously sea surface fluorescence. In order to validate the salinity measurements computed from conductimetry, discrete salinity samples were performed on a daily basis, before, during and after the campaign. They were measured on a PortaSal salinometer at the SHOM (Service Hydrographique et Oceanographique de la

Marine) with a precision of 0.002. A 1:1 relationship between TSG and analyzed salinity was obtained (R²=0.97, n=31) with a mean difference of 0.000 and a standard deviation of the residuals of 0.018. A total of 177 surface water samples were collected every 20 min from the TSG water outflow for the determination of nitrate, nitrite, phosphate and silicate concentrations (Sect. 2.3). Measurements for Chl-*a* (Sect. 2.3.) were collected randomly during day and night, leading to a total of 41 samples collected from the flow-through system. TSG fluorescence

signal was converted to Chl-*a* concentration values thanks to a comparison with Chl-*a* analysis evidencing a significant correlation between fluorescence and Chl-*a* with an R² of 0.50 (p-value<0.05). As Chl-*a* values obtained during OSCAHR were low (0.08 to 0.42 µg.dm$^{-3}$, with mean value of 0.15 µg.dm$^{-3}$), and considering the effect of fluorescence quenching, getting such a correlation was quite reasonable.

The CytoSense, an automated flow cytometer (AFCM) designed by the CytoBuoy, b.v. company (NL),

analyzed every 20 min samples isolated from the sea surface continuous flow-through system of the TSG. The



AFCM used in this study was specially designed to analyze the pulse shapes of phytoplankton size wide range (<1 – 800 µm in width and several mm in length) and abundance (within the ~0.5 cm$^3$ to the ~4.5 cm$^3$ analyzed). The analyzed seawater was pumped with a calibrated (weighing method) peristaltic pump from a discrete intermediate container, subsampling the continuous flow-through seawater into a 300 cm$^3$ volume to minimize

the spatial extent during the AFCM analyzing time. A sheath loop (NaCl solution (35‰) filtered on 0.2 µm) was used to separate, align and drive the particles to the light source and was continuously recycled using a set of two 0.1 µm filters (Mintech$^®$ fiber Flo 0.1 µm), completed with an additional carbon filter (PALL$^®$ Carbon filter) to reduce the background signal from the seawater and remove colloidal material. The sheath flow rate was 1.3 cm$^3$.s$^{-1}$. In the flow cell, each particle was intercepted by a laser beam (OBIS$^®$ laser, 488 nm, 150 mW) and the

generated optical pulse shape signals were recorded. The light scattered at 90° (sideward scatter, SWS) and fluorescence emissions were separated by a set of optical filters (SWS (488 nm), orange fluorescence (FLO, 552-652 nm) and red fluorescence (FLR, >652 nm)) and collected on photomultiplier tubes. The forward scatter (FWS) signal was collected onto two photodiodes to recover left and right signal of the pulse shape. Each particle passes at a speed of 2 m.s$^{-1}$ along the laser beam width (5 µm) with a data recording frequency of 4

MHz, generating optical pulse shapes used as a diagnostic tool to discriminate phytoplankton groups. Two distinct protocols were run sequentially every 20 min, the first one targeted autotrophic picophytoplankton with FLR trigger level fixed at 5 mV, sample flow rate at 5.0 mm$^3$.s$^{-1}$ for 3 min, resulting in ~ 0.5 cm$^3$ analyzed samples. The second dedicated to the analysis of nano- and microphytoplankton was triggered on FLR at 30 mV, sample flow rate was fixed at 10 mm$^3$.s$^{-1}$ for 10 min, resulting in ~4.5 cm$^3$ analyzed samples. Phytoplankton

groups were resolved using CytoClus$^®$ software generating several two-dimensional cytograms of retrieved information from the 4 pulse shapes curves (FWS, SWS, FLO, FLR) obtained for each single cell, mainly the area under the curve and the maximum of the pulse shape signal. Groups' abundances (cells.cm$^{-3}$), mean (a.u.cell$^{-1}$) and sum (product of mean properties per group abundances, a.u.cm$^{-3}$) of optical pulse shapes were processed with the software to assess their inherent dynamics. Up to 150 pictures of microphytoplankton were

collected during the FLR 30 mV acquisition by an image-in-flow camera mounted upward the flow cell. FWS scatter signals of silica beads (0.4, 1.0, 1.49, 2.01, 2.56, 3.13, 4.54, 5.02, 7.27 µm non functionalized silica microspheres Bangs Laboratories, Inc.) were used to convert light scatter to equivalent spherical diameter (ESD) and biovolume. A power law relationship (log(Size)=0.309*log(FWS)-1.853) allowed the conversion of the FWS signal into cell size (n = 17, R² = 0.94). The stability of the optical unit and the flow rates were checked

using Beckman Coulter Flowcheck™ fluorospheres (2 µm) before, during and after installation.





### 2.6 Vertical sampling

A Moving Vessel Profiler, MVP200 ODIM Brooke Ocean, equipped with a MSFFF I (Multi Sensor Free Fall Fish type I) containing an AML microCTD was deployed. The MVP casts were run from sea surface to 300 m depth during the vessel underway at a mean speed of 6 knots with continuous acquisition of temperature and salinity. Along most part of the campaign route, vertical profiles of temperature and salinity were obtained during the nearly vertical free fall with a temporal resolution of 8-10 min, corresponding to a spatial resolution of ~1 km. Salinity and temperature data acquired near the surface (~5 m) with the MVP were compared to the data acquired from the onboard TSG. MVP temperature and salinity values were significantly correlated to the continuous underway measurements with a 1:1 relationship, $R^2$ of 0.99 and 0.84 and root mean square error (RMSE) of residuals of 0.07°C and 0.02 for temperature and salinity, respectively.

A total of 8 fixed stations were performed (Fig. 2) and used to collect biogeochemical information and to validate the deployment of the MVP. For each station, a CTD-rosette cast down to 300 m recorded temperature, salinity and fluorescence profiles. At Station 11, the water column properties down to 1000 m was investigated with this CTD-rosette instrument. The CTD-rosette was equipped with a 12 Niskin bottle (12 $dm^3$) SBE32 Carousel water sampler and carried a CTD SBE911+ for temperature and salinity, a Chelsea Aquatracka III fluorimeter and a QCP-2350 (cosinus collector) for PAR measurements. Samples for nutrients and phytoplankton groups using bench top flow cytometry (Sect. 2.4.) were collected from the surface to 1000-m depth.

For stations 5 to 11 (Fig. 2), an innovative system of high-resolution seawater sampling down to 35 m (PASTIS_HVR – Pumping Advanced System To Investigate Seawater with High Vertical Resolution) was deployed. Seawater samples were collected using a Teflon pump (AstiPure™ II High Purity Bellows Pumps – Flow rate = 30 $dm^3.min^{-1}$) connected to a polyethylene (PE) tube fixed to the frame at the level of the pressure sensor of a Seabird SBE19+ CTD and a WetLab WETstar WS3S fluorimeter. The depth of the sampling was defined as the mean depth recorded by the pressure sensor with a vertical resolution of 0.1 to 1 m (depending on the sea state). The SBE19+ CTD offered precisions for temperature and computed salinity of 0.005°C and 0.002, respectively. The PASTIS_HVR was used to collect samples every 2-3 m for nutrients (Sect. 2.3.) and phytoplankton groups using bench top flow cytometry analyses (Sect. 2.4.). Random 27 seawater samples were collected and filtered to measure Chl-*a* concentration (Sect. 2.4.) and to convert fluorescence signal into Chl-*a* values. A significant correlation between fluorescence and Chl-*a* was obtained with an $R^2$ of 0.52 (p-value<0.05). A cross-calibration in terms of fluorescence was performed between fluorometers of the CTD-





Rosette and the CTD used for PASTIS_HVR to harmonize Chl-*a* values (fluorescence$_{CTD\_PASTIS\_HVR}$ = fluorescence$_{CTD\_rosette}$ x 3.31, n=60, R²=0.85).

**2.7 Surface specific growth rates and primary production estimates**

Phytoplankton growth rates were estimated by measuring independently with AFCM the net abundances combined with a size-structured population model described in Sosik et al. (2003) and adapted by Dugenne et al. (2014) and Dugenne (2017). Observed diel variations of single cell biovolumes within a specific cluster, retrieved from the power law relationship between cell size and FWS, were used as inputs for this size-structured population model. We identified the set of parameters that could optimally reproduce the diel variation of population size distribution using only cell cycle transitions by inverse modelling. In the model, temporal transitions of cells proportions in size classes, $\vec{v}$, are assumed to result from either cellular growth, supported by photosynthetic carbon assimilation, or asexual division. The increase of cell size occurring during the interphase is dependent of the proportions of cells that will grow between $t$ and $t + dt$, noted $\gamma(t)$. This probability is expressed as an asymptotic function of incident irradiance (Eq. (1)).

$$\gamma(t) = \gamma_{max}.\left[1 - exp\left(-\frac{Irradiance}{Irradiance^*}\right)\right] \tag{1}$$

with Irradiance : instantaneous PAR, Irradiance* : scaling parameter, $\gamma_{max}$ : maximal proportion of cells growing between $t$ and $t + dt$.

On the contrary, the decrease of cell size occurring after the mitosis marks the production of two daughter cells whose size has been divided by a factor 2. Thus the decrease of cell size is dependent on the proportion of cells that will enter mitosis between $t$ and $t + dt$, noted $\delta(t)$, which ultimately controls the population net growth rates (Eq. (2)).

$$\mu(t) = \frac{1}{dt}.ln(1 + \delta(t)) \tag{2}$$

Because natural populations show a clear temporal variation of the mitotic index ($\delta(t)$), the proportion of cells entering mitosis is expressed as a function of both time (Vaulot and Chisholm, 1987; André et al., 1999; Jacquet et al., 2001) and cell size (Maranon, 2015) (Eq. (3)).

$$\delta(t) = \delta_{max}.f(\mu_v\sigma_v^2).f(\mu_t\sigma_t^2) \tag{3}$$

with $f$ the Normal probability density, $v$ : cell size, $\delta_{max}$ : maximal proportion of cells entering mitosis, $\mu_v$ : mean of size density distribution, $\sigma_v$ : standard deviation of size density distribution, $\mu_t$ : mean of temporal density distribution, $\sigma_t$ : standard deviation of temporal density distribution.



By analogy with a Markovian process, the initial distribution of the cell size, $\vec{N}(0)$, is projected with a time step of $dt = \frac{10}{60}$ hour, to construct the normalized size distribution, $\vec{w}(t)$, over a 24h period (Eq. (4)).

$$\widehat{\vec{N}}(t + dt) = A(t).\widehat{\vec{N}}(t) \ \ and \ \ \widehat{\vec{w}}(t + dt) = \frac{A(t).\widehat{\vec{N}}(t)}{\sum A(t).\widehat{\vec{N}}(t)} \tag{4}$$

The tridiagonal transition matrix, $A(t)$ contains :

1. the stasis probability, expressed as the proportions of cells that neither grew nor divide between $t$ and $t + dt$

2. the growth probability ($\gamma$), expressed as the proportions of cells that grew between $t$ and $t + dt$

3. the division probability ($\delta$), expressed as the proportions of cells that entered division between $t$ and $t + dt$

The set of parameters, $\vec{\theta}$, is optimally identified (Eq. (5)) using the assumption of a Gaussian error distribution, $\sum(\vec{\theta})$ (Eq. (6)). Their standard deviation are estimated by a Markov Chain Monte Carlo approach

that sample $\vec{\theta}$ from their prior density distribution, obtained after running 200 optimizations on bootstrapped residuals, to approximate the parameter posterior distribution using the Normal likelihood.

$$\vec{\theta} = \{\gamma_{max}, Irradiance^*, \delta_{max}, \mu_v, \sigma_v, \mu_t, \sigma_t\} = argmin(\sum(\vec{\theta})) \tag{5}$$

$$\sum(\vec{\theta}) = \sum_{t=T0}^{T1\ day} \sum_{i=1}^{m}(\vec{w} - \widehat{\vec{w}}(\vec{\theta}))^2 \tag{6}$$

Ultimately, the equivalent of the temporal projection of proportions is conducted on the absolute diel

size distribution ($\vec{N}$) with the optimal set of parameters to estimate population intrinsic growth rates ($\mu$) on a 24 h period, from which the hourly logarithmic difference of observed abundances is subtracted to obtain the daily average population losses rates ($\bar{l}$) (Eq. (7)).

$$\mu = \frac{1}{24.\frac{1}{dt}+1}.ln\left(\frac{\widehat{\vec{N}}(T_{1\ day})}{\vec{N}(T_0)}\right) \text{ and } \bar{l} = \int^{dt=1h} \mu(dt) - \frac{1}{dt}.ln\left(\frac{\vec{N}(t+dt)}{\vec{N}(t)}\right) \tag{7}$$

The ratio between mean cell biovolume at dawn (min) and dusk (max) has been used for *Synechococcus*

and other phytoplankton groups (Binder et al., 1996; Vaulot and Marie, 1999) as a minimum estimate of the daily growth rate. This simple approach assumes that cell growth and division are separated in time (synchronous population), whereas these processes occur simultaneously in a population (Waterbury et al., 1986; Binder and Chisholm, 1995; Jacquet et al., 2001). Since the model allows for any cell to grow, divide or be at equilibrium over the entire integration period (asynchronous populations), growth rates $\mu_{size}$ superior to the

median size ratio $\mu_{ratio} = ln(\bar{v}_{max}/\bar{v}_{min})$ (indicative of a synchronous population), are assumed to be well represented.

The apparent increase of carbon biomass, defined as the Net Primary Production $NPP_{cell}$ (Eq. (8), mg C.m$^{-3}$.d$^{-1}$), was calculated using a constant cell to carbon conversion factor $Q_{C, calc.}$ (Table 2).



$$NPP_{cell} = Q_c . \delta(t) . \widehat{\overline{N}}(T_0) = Q_c . [\exp(\mu(t)) - 1] . \widehat{\overline{N}}(T_0) \qquad (8)$$

The biovolume to carbon $av_i^b$ relationship (Table 2), was used to calculate the Net Primary Production $NPP_{size}$ (Eq. (9)) in a different way:

$$NPP_{size} = \sum_{t \in R^*} \Delta(< C_{size}, \widehat{\overline{N}}(t + dt) >, < C_{size}, \widehat{\overline{N}}(t) >)$$
$$= \sum_{t=T_0}^{T_0 + 1 \, day - dt} \sum_{i=1}^m av_i^b (N_i(t + dt) - N_i(t)) \qquad (9)$$

These conversions allow approximating the daily NPP using the approximation of the carbon content of the cells newly-formed after mitotic division over 24 hours ($NPP_{cell}$), or directly assimilated by photosynthesis during the photoperiod ($NPP_{size}$). The estimations result from the apparent mitotic index optimally deduced from the diel dynamics of the normalized size distribution. They do not accommodate any cells removal processes within the period of integration that could be caused by grazing or physical transport.

## 3 Results

### 3.1 Description of the fine-scale structure

Surface currents distributed by AVISO exhibited a cyclonic recirculation in the Ligurian subbasin (Fig. 1). Current velocities and directions measured by the ADCP were in general agreement with the altimetry derived ones. The highest current velocities (> 0.3 m.s$^{-1}$) were associated with the Northern Current. The main cyclonic circulation was divided in two parts: a small recirculation centered on (8.75°W, 43.80°N) and a second one in the southwest separated by a local minimum in current intensity, both observed in AVISO and ADCP data.

Between the 30/10 and the 2/11, a strong North-East wind event (wind velocity up to 70 km.h$^{-1}$) was recorded all over the area, associated with a SST drop of ~1°C in the Ligurian subbasin. Satellite SST images from the 30/10 to the 6/11 (Fig. 1) evidenced a patch of cold surface waters with values below 17.5°C. The observation was confirmed by the ship surface TSG between the 3/11 and the 6/11 (mean SST of 16.3 ± 0.3°C and mean SSS below 38.20 (Fig. 2, Table 1)). The cold patch was surrounded by warmer surface waters with SST up to 19°C, validated by *in-situ* records from the TSG. Both satellite and *in-situ* sampling described two warm boundaries (see Sect. 4.2.) characterized by SST higher than 17.5°C and SSS above 38.20. The lowest SST values were observed between the 3/11 and the 5/11, then the patch warmed up on the 6/11. Remotely-sensed SST were well correlated with the one recorded by the TSG along the ship track (R²=0.82, p-value<0.05), even if





remote-sensing tended to underestimated SST. Temperature gradients observed from the TSG were well caught by satellite products.

Figure 3 depicts temperature and salinity vertical section of a south-to-north MVP transect from 00:00 to 06:00 (local time) on the 5/11. The thermocline was located between 20 m – 30 m depth in cold core area and

between 30 m – 40 m abroad. Temperatures above the thermocline were uniform in the cold core and warm boundaries waters, while within the transition areas temperatures increased progressively from the thermocline to the surface (Fig. 3 and Fig. S1 in supplementary material). The deep water temperature, below the thermocline, ranged from 13.5°C to 14.5°C and did not present any significant differences between the cold core and the warm boundaries. Sea surface salinity (SSS) were lower (<38.20) in the cold core than in warm

boundaries (>38.20) and salinity at 300 m depth was higher than 38.50. A subsurface layer of low-salinity waters (<38.10) spread off below the thermocline with a 40 m to 80 m thickness. This subsurface layer was observed up to the surface in the center of the cold core, whereas in warm boundaries saltier (>35.20) surface waters overlaid it.

Remotely-sensed Chl-*a* concentration estimates ranged between 0.10 and 0.30 µg.dm$^{-3}$ during the

campaign (Fig. 1). Unfortunately cloud cover masked the remote-sensing Chl-*a* from the 3/11 to the 5/11. The study area (black square on Fig. 1) was considered as case 1 waters. The 30/10, remotely-sensed surface Chl-*a* concentrations ranged from 0.10 to 0.20 µg.dm$^{-3}$. The 2/11, concentrations higher than 0.30 µg.dm$^{-3}$ were observed in the center of the cold patch and decreased below 0.20 µg.dm$^{-3}$ on the 6/11. Mean satellite Chl-*a* estimates recorded and averaged from the 2/11 to the 6/11 were significantly correlated with Chl-*a* derived from

the ship fluorometer during the campaign (R²=0.47, p-value<0.05). The highest Chl-*a* concentrations measured from TSG fluorescence were recorded in the center of the cold patch, with Chl-*a* concentrations up to 0.40 µg.dm$^{-3}$ and mean Chl-*a* of 0.17 ± 0.04 µg.dm$^{-3}$ (Table 1), while warm boundaries presented lower Chl-*a* concentrations (< 0.15 µg.dm$^{-3}$).

Surface nutrient variability was investigated from the 177 discrete samplings performed every 20

minutes (Table 1). Surface nitrate, nitrite and phosphate concentrations were below or close to the detection limits (< 0.05 µmol.dm$^{-3}$) excluding any spatial variability observation. Only silicate concentration presented variability in its distribution with mean values of 1.31 ± 0.05 µmol.dm$^{-3}$ in the cold core and 1.19 ± 0.06 µmol.dm$^{-3}$ in the warm boundary surface water (Table 1).

Deep Chl-*a* maxima (DCM) were observed at the vicinity of 30 m and 45 m depth for the cold core and

warm boundaries stations, respectively (Fig. S1 in supplements). DCM occurred approximately 10 m below the thermocline. DCM concentrations were characterized by Chl-*a* values between 0.30 µg.dm$^{-3}$ and 0.40 µg.dm$^{-3}$ in





the cold core and 0.20 µg.dm$^{-3}$ and 0.30 µg.dm$^{-3}$ in the warm boundaries waters. The euphotic zone ($Z_{eu}$) spread down around 70 m all over the study area (Fig. S1 and S2).

**3.2 Phytoplankton groups definition**

Up to ten phytoplankton groups were resolved by AFCM on the basis of their light scatter (namely
forward scatter FWS and sideward scatter SWS) and fluorescence (red FLR and orange FLO fluorescence ranges) properties over the 177 validated samples collected using two-dimensional projections (cytograms, Fig. 4). Two main groups, *Prochlorococcus* and *Synechococcus*, were optimally resolved using a low FLR trigger level (FLR 5 mV) and adequately counted within ~0.5 cm$^3$ of sample analyzed. Using this configuration, picoeukaryotes, nanoeukaryotes and microeukaryotes could also be observed, but to get more accurate
abundances of these less abundant microorganisms, a second analysis with a larger volume analyzed (5 cm$^3$) was performed with a higher trigger level (FLR 30 mV) in order to not take into consideration the smallest and most abundant cells (*Prochlorococcus* for instance). Due to their small sizes and their limited photosynthetic pigment contents, *Prochlorococcus* were resolved close to the limit of the AFCM detection capacity means of the maximum SWS and FLR pulse shape curves. Cells assigned to *Synechococcus* group were unambiguously
resolved thanks to their higher FLO intensity compared to their FLR intensity (Fig. 4a, b) induced by the presence of phycoerythrin pigments. According to a log-log linear regression relying FWS to the equivalent spherical diameter (ESD), *Prochlorococcus* and *Synechococcus* exhibited an median ESD of 0.5 ± 0.1 µm (0.07 ± 0.03 µm$^3$) and 0.9 ± 0.2 µm (0.46 ± 0.38 µm$^3$), respectively (Table 2). *Prochlorococcus* and *Synechococcus* continuous surface counts were compared to conventional flow cytometry analysis performed with the
FACSCalibur flow cytometer on discrete samples collected at fixed stations at the 2 first sampling depths near the surface (Fig. S3). Both methods counts did not show significant differences (t-test, p-value<0.001), which validates the observations obtained with the automated CytoSense. A post-campaign validation of *Prochlorococcu*s, *Synechococcus* and standard beads (2µm *Polyscience* yellow/green beads) counts was performed by analyzing 5 natural seawater replicates on both the CytoSense and the FACSCalibur. Both
cytometers led to similar counts (Student test p>0.4, Table S1 in Supplements).

With higher trigger level (FLR30) it was possible to resolve and count larger cells in 5 cm$^3$, from picoeukaryotes to microeukaryotes (Fig. 4c, d). Three groups of picoeukaryotes were resolved on the basis of their optical properties. The main picoeukaryote group (PicoE) exhibited higher FLR and FWS and lower FLO intensities than *Synechococcus* with an ESD of 2.6 ± 0.5 µm (10.5 ± 5.5 µm$^3$) (Table 2). Picoeukaryotes groups
with high FLO (PicoHighFLO) and other with high FLR (PicoHighFLR) were also identified during the



campaign (Fig. 4c, d). Three distinct nanoeukaryote groups were defined according to their red and orange fluorescence properties. The main nanoeukaryote group (NanoE) had a FLR/FLO ratio close to PicoE ratio (Fig. 4c) with an ESD of $4.1 \pm 0.5$ µm ($37.0 \pm 14.7$ µm$^3$) (Table 2). Nanoeukaryotes cells, which emitted orange fluorescence with higher intensities than red fluorescence, were divided in two additional groups: NanoFLO and

NanoHighFLO, respectively. The distinction between nano- and microeukaryotes was done by combining FWS and the pictures collected by the image-in-flow device of the CytoSense. Two types of microeukaryotes were distinguished: microeukaryotes (MicroE) with size ranging between 10 and 20 µm and microeukaryotes with high FLO (MicroHighFLO) with size above 20 µm. The relatively small size of most of the MicroE limited their identification. The MicroE was not properly a microphytoplankton group according to the official size

classification (20 - 200 µm) but it was distinct from the 3 nanoeukaryote groups (ESD < 5 µm).

### 3.3 Phytoplankton groups distribution

Figure 5 shows the surface abundance of *Prochlorococcus*, *Synechococcus*, picoeukaryote and nanoeukaryote groups over the study area. Picoeukaryote and nanoeukaryote abundances were computed as the sum of the three picoeukaryote (PicoE, PicoHighFLO and PicoHighFLR) and nanoeukaryotes (NanoE,

NanoFLO and NanoHighFLO) groups, respectively, in order to simplify the representation of the phytoplankton group distribution. *Prochlorococcus* abundances varied between 8,800 and 51,500 cells.cm$^{-3}$ (Fig. 5a) with higher abundances in the center of the structure (> 30,000 cells.cm$^{-3}$), corresponding to the cold core (Fig. 2a). In warm boundaries, *Prochlorococcus* abundances were below 30,000 cells.cm$^{-3}$, with on average $20,000 \pm 6,000$ cells.cm$^{-3}$ (Table 1). *Synechococcus* population ranged from 13,500 to 35,900 cells.cm$^{-3}$ (Fig. 5b). In the patch of

cold waters, *Synechococcus* mean abundance was $18,000 \pm 3,000$ cells.cm$^{-3}$ and in surrounding warm waters a mean abundance of $25,000 \pm 3,000$ cells.cm$^{-3}$ was observed. Picoeukaryote abundances varied between 875 and 2,040 cells.cm$^{-3}$ and nanoeukaryote abundances ranged from 567 to 1,175 cells.cm$^{-3}$. Picoeukaryote and nanoeukaryote populations presented a similar surface distribution pattern as the *Prochlorococcus* one, with higher abundances in the cold core than in warm boundaries. In the cold patch, mean abundances of $1,200 \pm 200$

cells.cm$^{-3}$ and $890 \pm 90$ cells.cm$^{-3}$ were observed for picoeukaryotes and nanoeukaryotes, respectively. Warm boundary surface waters hosted picoeukaryote and nanoeukaryotes average populations of $900 \pm 100$ cells.cm$^{-3}$ and $780 \pm 130$ cells.cm$^{-3}$, respectively (Table 1). PicoHighFLO and NanoFLO did not exhibit clear pattern between cold core and warm boundaries (data not shown) with abundances varying between 50 and 150 cells.cm$^{-3}$. PicoHighFLR abundance were below 100 cells.cm$^{-3}$ during all the campaign, excepted at the vicinity

of the station 8 (Fig. 2), where they reached up to 400 cells.cm$^{-3}$ and where the highest Chl-*a* values were



recorded (Fig. 6). NanoHighFLO showed the same behavior as PicoHighFLR with abundances below 50 cells.cm$^{-3}$ during the campaign and a peak up to 200 cell.cm$^{-3}$ in the same area. Variations of microeukaryotes (between 20 and 30 cells.cm$^{-3}$ and bellow 5 cells.cm$^{-3}$ for MicroE and MicroHighFLO, respectively) are not shown considering their low and relatively homogeneous abundances during the campaign and throughout the different type of surface waters (Table 1). Although MicroHighFLO abundances were exceptionally high at the vicinity of station 8 (up to 20 cell.cm$^{-3}$).

Figure 6 illustrates the temporal surface variability of *Prochlorococcus*, *Synechococcus*, picoeukaryote and nanoeukaryote abundances together with temporal variations of SST, SSS and Chl-*a* concentration. *Prochlorococcus* and *Synechococcus* abundances exhibited an opposite distribution throughout the cold and warm surface waters, with dominance of *Prochlorococcus* in cold core waters and of *Synechococcus* in warm boundary waters. These shifts fitted perfectly with the short terms transitions observed from the SST all along the cruise. Picoeukaryote maximal abundances (around 2,000 cells.cm$^{-3}$) were observed simultaneously with the highest values of Chl-*a* concentrations in cold waters and lower abundances were found out in warm and Chl-*a* poor surface waters. Nanoeukaryote population followed a similar trend.

### 3.4 Contribution to total fluorescence and carbon biomass

The relative contribution to red fluorescence FLR$_i$ of *Prochlorococcus*, *Synechococcus*, picoeukaryotes (PicoE, PicoHighFLO, PicoHighFLR), nanoeukaryotes (NanoE, NanoFLO, NanoHighFLO), and microphytoplankton (MicroE, MicroHighFLO) groups were obtained by multiplying their mean cell's red fluorescence intensity (FLR$_m$) recorded by AFCM by their respective abundances according to $FLR_i = (FLR_{m,i} * Abundance_i)$. The integrative FLR$_{Total}$ signal was calculated as $FLR_{Total} = \sum_i FLR_i$. The ratios FLR$_i$/FLR$_{Total}$ give an estimate of the contribution of each phytoplankton group to the bulk fluorescence signal. A significant correlation (R²=0.80, n=144) was established between computed FLR$_{Total}$ and Chl-*a* concentrations derived from continuous surface fluorescence measurements (Fig. 7b, by excluding the orange dots). When one considers only the relative contributions of each group in the cold core (blue) and warm boundary of type 1 (red), *Prochlorococcus* contributed to 4.4 ± 1.7% and 2.5 ± 1.1% of FLR$_{Total}$, *Synechococcus* FLR accounted for 24.5 ± 4.2% and 33.3 ± 4.4% of FLR$_{Total}$, respectively. Picoeukaryotes contributed to 14.4 ± 1.9% and 11.7 ± 1.9% of FLR$_{Total}$ and nanoeukaryotes FLR accounted for 50.6 ± 4.5% and 46.5 ± 6.1% of FLR$_{Total}$, in cold core and warm boundary 1 respectively. Microphytoplankton contribution was around 6% in both hydrographical provinces, with a peak of contribution (>10%) observed at the vicinity of the highest Chl-*a* values recorded near station 8 (Fig. 7a).



Similar calculation of C biomass were performed according to the cellular C quota ($Q_{C, calc.}$, Table 2) defined for *Prochlorococcus*, *Synechococcus*, picoeukaryote and nanoeukaryote groups and their abundances. The C individual cellular quota ($Q_{C, calc.}$) has been derived from the average cell size according to the allometric regression formula $Q_{C, calc.} = a.Biovolume^b$ (Menden-Deuer and Lessard (2000)). This yielded to average C

biomasses of 25 fgC.cell$^{-1}$ for *Prochlorococcus* cells, 109 fgC.cell$^{-1}$ for *Synechococcus* cells, 1880 fgC.cell$^{-1}$ for picoeukaryote cells and 9000 fgC.cell$^{-1}$ for nanoeukaryote cells (Table 2). No cellular C quota was assigned to the microphytoplankton cluster regarding the large size range observed (from 10 µm up to 80 µm). In cold core waters, the relative contribution of *Prochlorococcus*, *Synechococcus*, picoeukaryote and nanoeukaryote groups were 6 ± 1%, 14 ± 2%, 20 ± 2% and 60 ± 3%, respectively. In warm boundary waters, these relative

contributions accounted for 4 ± 1%, 22 ± 3%, 17 ± 2% and 57 ± 5%. FLR and C biomass followed the same dynamics between both hydrographical provinces.

**3.5 Fine-scale vertical variability**

The fine-scale vertical variability of temperature, salinity and Chl-*a* concentration was investigated in the first 35 m of the water column during several discrete station stops together with phytoplankton abundances

sampled every 2-3 m (Fig. 8) with the dedicated PASTIS_HVR pump system. Complementary nutrient analyses were made at a lower vertical resolution (10 m). Nitrite and phosphate concentration profiles never overpassed the limits of quantification of the analyzers (data not shown). Fixed stations were grouped in cold core (stations 5, 8, 9, 11) and warm boundary stations (stations 6, 7, 10) depending on their surface water temperatures (Fig. 2, Fig. 6). Profiles performed at warm boundary stations over the first 35 m were mostly homogeneous.

Temperatures ranged between 18°C and 19°C, salinity values were higher than 38.20 and Chl-*a* concentrations were lower than 0.10 µg.dm$^{-3}$. Nitrate concentrations remained lower than 0.05 µmol.dm$^{-3}$ and silicate concentrations varied between 1.15 and 1.20 µmol.dm$^{-3}$. Picophytoplankton abundances exhibited the same uniform vertical patterns. *Prochlorococcus* abundances remained below 30,000 cells.cm$^{-3}$, *Synechococcus* population counted over 30,000 cells.cm$^{-3}$ and picoeukaryotes varied between 800 and 1,200 cells.cm$^{-3}$. As

previously described in Sect. 3.1, the thermocline was located between 30 m and 40 m, below the PASTIS_HVR sampling depth. Profiles performed in cold surface waters area showed a decrease of temperatures from 15 m to 30 m depth occurring together with an increase of Chl-*a* concentrations up to 0.60 µg.dm$^{-3}$. Higher values of nitrate and silicate were recorded concomitantly with the temperature drawdown and Chl-*a* increase. *Prochlorococcus* and picoeukaryotes populations became more abundant in depth, and reached concentrations

up to 97,000 cells.cm$^{-3}$ and 5,200 cells.cm$^{-3}$, while *Synechococcus* abundance tended to decrease, together with



the temperatures, below 4,000 cells.cm$^{-3}$. Station 11 (Fig. 2) was considered as a cold core station regarding its vertical profile (Fig. 7) even though surface was relatively warm (Fig. 6) with *Synechococcus* more abundant than *Prochlorococcus*. Station 11 was positioned in a transition area between the warm boundaries and the cold core. In the cold core stations, vertical profiles exhibited heterogeneous patterns, because of a shallower

thermocline (Fig. S1), impacting physical and biogeochemical fine-scale variability. These results corroborated the observations obtained from the MVP profiles (Fig. 3), suggesting a shallowing of the thermocline and the associated surface mixed layer limit in the cold core.

**3.6 Growth rates and primary production estimates**

*Prochlorococcus* and *Synechococcus* size distributions were retrieved over 24h according to the power
law function relying FWS to biovolume. The 5/11, hourly measurements of individual cell FWS obtained in warm boundary waters (Fig. 6) were used to follow the diurnal variability of population size distribution in warm boundary waters although the sampling frequency spanned 20 min (Fig. 6). This timeframe was selected to avoid the consideration of diel measurements collected in both cold core and warm boundary waters, which presented different physical and biogeochemical properties. The 5/11, cold core waters were crossed several times by the
ship within the 1h-timeframe. Figures 9a and 9b show the hourly cell biovolume variations over 24h for *Prochlorococcus* and *Synechococcus*, respectively. A diurnal cycle was described for both populations with minimal and maximal biovolumes observed at 6:00 and at 18:00 (local time), respectively. *Prochlorococcus* biovolume varied from 0.04 µm$^3$ (ESD = 0.42 µm) to 0.12 µm$^3$ (ESD = 0.61 µm) between dawn and dusk. At the end of the dark period (6:00), *Synechocococcus* biovolume decreased down to 0.20 µm$^3$ (ESD = 0.72 µm) and at
the end of the photoperiod, biovolume reached values up to 0.60 µm$^3$ (ESD = 1.04 µm). The size distribution variations observed for both populations, with a clear diurnal cycle pattern, highlighted the capacity of single cell flow cytometry measurements to follow the cellular cycle of these picophytoplanktonic populations. Similar computations were performed on pico- and nanoeukaryote population but their size distribution did not show pattern consistent with the assumption of the size distribution model.

Using a size-structured matrix population model, *in-situ* daily growth rates were estimated from the predicted absolute distribution of cell in size classes, with the continuously observed size distribution as model input. *Prochlorococcus* and *Synechococcus* modeled–produced cell size distribution (Fig. 9c and 9d) reproduced well the diurnal size distribution cycle and allowed us to derived specific growth rate ($\mu_{size}$, Table 3) for both populations. For comparison, the median size ratio $\mu_{ratio} = ln(\bar{v}_{max}/\bar{v}_{min})$ (Table 3) was computed.
*Prochlorococcus* and *Synechococcus* specific growth rates $\mu_{size}$ were 0.21 ± 0.01 d$^{-1}$ and 0.72 ± 0.01 d$^{-1}$; 0.28 and



0.49 for mean size ratio $\mu_{ratio}$, respectively. *Prochlorococcus* computed loss rate estimate was 0.30 d$^{-1}$, while *Synechococcu*s was characterized by a computed loss rate of 0.68 d$^{-1}$.

The apparent production of these picocyanobacteria NPP$_{cell}$ and NPP$_{size}$ were computed from the populations intrinsic growth rates (Eq. (8) and (9)), in the absence of particles grazing and sinking and of

advective processes, using the approximation of the carbon content Q$_{C, calc.}$ (Table 2) of the cells newly formed after mitotic division over 24 hours. *Prochlorococcus* NPP$_{cell}$ was 0.11 mgC.m$^{-3}$.d$^{-1}$ and *Synechococcus* NPP$_{cell}$ estimate was 2.68 mgC.m$^{-3}$.d$^{-1}$ (Table 3) considering mean carbon cellular quota of 25 and 109 fgC.cell$^{-1}$ for *Prochlorococcus* and *Synechococcus* (Table2). Accounting for the increase of their size distribution during the photoperiod, *Prochlorococcus* NPP$_{size}$ was estimated at 0.13 mgC.m$^{-3}$.d$^{-1}$ and *Synechococcus* NPP$_{size}$ estimated at

2.80 mgC.m$^{-3}$.d$^{-1}$ (Table 3) using biovolume to carbon $av_i^b$ relationship for *Prochlorococcus* and *Synechococcus* (Table2).

## 4 Discussion

### 4.1 Physical origins and dynamics of the fine-scale structure investigated during OSCAHR

Both ADCP and AVISO derived surface currents directions and intensities suggested that the sampled

cold core mesoscale structure was associated to a cyclonic gyre generated by a recirculation of the Northern Currents towards the Western Corsican Current (Fig. 1, AVISO). Besides a generally cyclonic circulation pattern between the French coast and Corsica that geostrophically domed the isopycnals, Ekman pumping is likely to have played an important role since strong wind events were observed before the OSCAHR cruise and previous studies (Gaube et al., 2013) have highlighted Ekman pumping's impact on ocean biogeochemistry. Ekman

pumping calculated using both WRF and scatterometer wind estimates (Fig. 10) suggested that, besides the strong wind event occurring during the first day of the cruise, the region has experienced several wind events two weeks before the cruise characterized by vertical velocities peaking to 3-4 m.d$^{-1}$ inducing a strong decline in SST. Furthermore the time series of vertical velocities highlighted that the cold water "patch" experienced almost constantly negative (i.e. upwarding) vertical velocities for about one month (Fig. 10).

The shallowing of the thermocline in the central part of the cyclonic structure associated with low SST in the cold patch was evidenced by the MVP salinity and temperature profiles (Fig. 3). Low salinity waters at the surface of the cold patch supports the Ekman pumping process hypothesis. Within the warm boundaries, a subsurface layer of low-salinity waters (<38.10) spread off below the thermocline and reached the surface in the cold core, are observed for each MVP and CTD deployment. The origin of these low salinity waters remains



unclear. The cyclonic circulation in the Ligurian subbasin induced by the intense coastal currents along Italian and French coasts (Astraldi et al., 1994) is supposed to isolate the central Ligurian subbasin from direct riverine inputs, such inputs being in addition particularly poor in this area (Migon, 1993). Goutx et al. (2008) reported similar observations at the same period (13[th] of October 2004) in the Ligurian subbasin (43.25°N, 8°E, 48 km

offshore), close to our study area, as well as Marty et al. (2008). Further investigations might be done to find out the origin of this low saline subsurface layer.

As mentioned by McGillicuddy (2016), the superposition of a wind-driven Ekman flow on a mesoscale velocity field can cause ageostrophic circulation involving significant vertical transport (Niiler 1969, Stern 1965). The cyclonic recirculation produced a zone of divergence in the central zone of the Ligurian Sea which

domed the main pycnoclines, thereby shallowing the mixed layer (Sournia et al., 1990; Estrada, 1996; Nezlin et al., 2004). This process resulted in the fertilization of the upper mixed layer with nutrient rich upwelled waters (Miquel et al., 2011). Remote-sensing (SST, Chl-*a*), model (AVISO, WRF), continuous surface measurements and MVP profiles support the Ekman pumping hypothesis induced by a strong wind event. The resulting upwelled subsurface cold water fertilized surface waters, which increased Chl-*a* concentration (Fig. 1, 2 and 6)

and the primary production (Sournia et al., 1990) that, in turn sustain higher trophic levels (Warren et al., 2004).

Furthermore, surface warm boundaries waters were sub-divided in two distinct types (Table 1): type 1 (in red in Fig. 6, 10 and 11) and type 2 (in orange), according to their physical and biogeochemical properties. Cold patch waters (Fig. 7d) signature had SST lower than 17.5°C and SSS lower than 38.23. Type 1 warm boundary waters were defined with SST higher than 17.5°C and SSS higher than 38.23. Type 2 warm boundary

waters were characterized by SST higher than 18°C and SSS lower than 38.24.

### 4.2 Nutrients and Chl-*a* distribution

In the cold core, nitrate and silicate started to increase below 15 m (Fig. 8). The first detectable phosphate concentrations appeared below 50m (> 0.2 µmol.dm$^{-3}$, Fig. S2). However, surface cold core waters contained more autotrophic biomass than warm boundary waters as shown by surface Chl-*a* concentrations (Fig.

2 and 6, Table 1). In the cold core waters, the nutrient availability starting around 15-20 m depth sustained an increase in Chl-*a* up to 0.6 µg.dm$^{-3}$ at 30 m depth (Fig. 8), while in warm boundary waters, a deeper MLD maintained the DCM below 30m (Fig. S1). This later was characterized by lower Chl-*a* values in the warm boundary either limited by the nutrient availability and the amount of light availability for phytoplankton cells. Within the Ligurian subbasin, the DCM is shallower than in other oligotrophic areas: maximum of 60 m depth

(Marty et al., 2002) against 150m or more in the tropical oligotrophic Pacific ocean (Claustre et al., 1999),



~100m in the oligotrophic Atlantic gyres (Maranon et al., 2003) The euphotic depth (Zeu ~70m, Fig. S2 in supplements) in the Ligurian subbasin was deeper than the MLD and the DCM during all the year (Marty et al., 2002), excepted in winter. The variation of the nitracline depth induced by the cyclonic circulation and Ekman pumping appeared as the most relevant factor controlling this vertical and horizontal biological distribution
variability.

### 4.3 High-resolution dynamics of phytoplankton groups

### 4.3.1 Phytoplankton functional group description

The picocyanobacteria *Prochlorococcus* and *Synechococcus* are the smallest and most abundant photoautotroph in the oceans (Waterbury et al., 1986; Olson et al., 1988; Chisholm et al., 1992) and have a key
role in a variety of ecosystems, particularly in oligotrophic ones (Partensky et al, 1999a). *Synechococcus* are easily detectable by flow cytometry due to the bright orange fluorescence emitted by phycoerythrin during the excitation by the blue 488 nm laser beam of the flow cytometer. *Prochlorococcus*, which are smaller than *Synechococcus*, are characterized by very dim red fluorescence induced by Chl-*a*. The observations reported in this study are, to the best of our knowledge, the first to correctly resolved *Prochlorococcus* abundance in surface
waters using a CytoSense AFCM due to some improvements of the instruments (a carbon activated filter to reduce the optical background of the seawater, a more powerful laser beam to improve the side scatter intensities of these very small cells). *Prochlorococcus* mean ESD and associated biovolume of $0.5 \pm 0.1$ µm and $0.07 \pm 0.03$ µm$^3$, respectively (Table 2), were in the lower range of 0.5 to 0.9 µm and 0.03 to 0.38 µm$^3$ ESD and biovolume values reported in previous studies (Morel et al. 1993; Partensky et al., 1999b; Shalapyonok et al.,
2001; Ribalet et al., 2015). Sieracki et al. (1994), DuRand et al. (2001) and Shalapyonok et al. (2001) noticed that *Prochlorococcus* cell diameter and biovolume were generally lower in the surface mixed layer ($0.45 - 0.60$ µm and $0.05 - 0.11$ µm$^3$) than in deeper waters ($0.75 - 0.94$ µm and $0.21 - 0.43$µm$^3$). In this study, *Synechococcus* mean ESD and associated biovolume of $0.9 \pm 0.2$ µm, $0.46 \pm 0.38$ µm$^3$, respectively (Table 2), were in the same range of 0.8 to 1.2 µm and 0.25 to 1.00 µm$^3$ as ESD and biovolume values reported in previous
studies (Morel et al. 1993; Shalapyonok et al., 2001; Sosik et al., 2003; Hunter-Cevera et al., 2014). DuRand et al. (2001) and Shalapyonok et al. (2001) reported that deepen *Synechococcus* can also be characterized by higher mean cell diameters. To explain our observations, literature reveals that *Prochlorococcus* can belong to the photoadapted high-light HL ecotype characterized with less Chl-*a* content, i.e. less FLR, or to the low-light (LL) ecotype characterized with higher Chl-*a* content, i.e. higher FLR (Moore and Chisholm, 1999; Garczarek et al.,
2007; Partensky and Garczarek, 2010). Usually, the HL ecotype occupies the upper part of the euphotic zone,





while the LL ecotype dominates the bottom of the euphotic layer. *Synechococcus* ecotypes distribution is not characterized by a clear depth partitioning; their distribution appears to be principally controlled by water temperature and latitude (Pittera et al., 2014). Mella-Flores et al. (2011) and Farrant et al. (2016) reported that in Mediterranean Sea HLI and III clades were the dominant ecotypes surface waters for *Prochlorococcus* and

*Synechococcus*, respectively, whereas LLI and I/IV clades were the main *Prochlorococcus* and *Synechococcus* ecotypes present in deep waters. Obviously, further analyzes of OSCAHR samples performed at the molecular level would have been necessary to validate or not these explanations.

Pico- and nanoeukaryotes were distinguished in six cytometric groups based on their scattering (FWS) and fluorescence (FLR and FLO) properties, although pico- and nanoeukaryotes include cells of several taxa

(Simon et al., 1994; Worden and Not, 2008; Percopo et al., 2011). As mentioned in Sect. 3.3, PicoE and NanoE (Fig. 4) were the main groups represented in terms of abundances, and their variability drove the whole dynamics of pico- and nanoeukaryote size groups across the cold core and warm boundary waters. At the vicinity of station 8, where Chl-*a* concentrations up to 0.40 µg.dm$^{-3}$ were recorded, maxima (in terms of abundance) of PicoHighFLR, NanoHighFLO and MicroHighFLO were observed (data not shown, Sect. 3.3.).

PicoE group was characterized by a mean ESD of 2.6 ± 0.5 µm with a mean C quota of 1880 fgC.cell$^{-1}$ (Table 2). If flow cytometry is ataxonomic, it has been reported in several previous studies that picoeukaryote size fraction in the Mediterranean Sea are represented by radiolarians, alveolates, dinoflagellates and stramenopiles (Not et al., 2009), in the size spectrum 0.9 µm (*Ostreococcus taurii*) – 3.5 µm (*Phaeocystis cordata*). A global compilation from Vaulot et al. (2008) reported picoeukaryote description in an extended range of 0.8-3 µm,

which corresponds to the mean size over 2 µm observed in our study.

The mean ESD of the main nanoeukaryote functional group observed, NanoE (Fig. 4), was 4.1 ± 0.5 µm (Table 2), a relatively small size considering the 2-20 µm range characterizing nanoeukaryotes in the literature. The NanoHighFLO functional group (Fig. 4), characterized by high orange fluorescence presented similarities with the well-defined cryptophycea taxa, diagnosed by the presence of orange fluorescing

phycoerythrin. NanoHighFLO cells had mean ESD lower than 5 µm. In North-Western Mediterranean Sea, according to the abundant literature, the 2-10 µm size fraction is composed, in importance, of diverse genera of Coscinodiscophyceae (*Arcocellulus* : 3.5-8.7 µm, *Minidiscus* : 2.7-4.3 µm, *Thalassiosira* : 2.7-16.3 µm), Dinophyceae (*Heterocaspa* : 7.0-10.6 µm), Coccolithophyceae (*Anthosphaera* : 2.9 µm, *Gephytocapsa* : 4.7-8.3 µm), Prymnsiophyceae (*Chrysochromulina* : 3.2-4.0 µm) (Percopo et al., 2011). Small nanoflagelattes dominate

the nanophytoplankton size group in terms of cell concentrations most of the year in oligotrophic Mediterranean Sea waters (Siokou-Frangou et al., 2010).



Microphytoplankton abundances reported in this study (20-30 cell.cm$^{-3}$) could appear high, regarding previously reported cell concentrations ranging between 1 and 5 cells.cm$^{-3}$ (Gomez and Gorsky, 2003). MicroE cells, as defined manually on cytograms, presented ESDs comprised between 10 and 20 μm, which could be considered as large size nanophytoplankton cells. As mentioned by Siokou-Frangou et al. (2010), single cells of

colonial diatoms smaller than 20 μm are commonly observed in Mediterranean waters and are treated separately than the nanophytoplankton because of their larger functional size and distinct ecological role. MicroHighFLO cluster had mean ESD > 20 μm, and were considered as the only true microphytoplankton component. MicroHighFLO abundances (< 5 cells.cm$^{-3}$, with a peak up to 20 cells.cm$^{-3}$) were in better agreement with those generally observed in similar oligotrophic surface waters (Gomez and Gorsky, 2003; Vaillancourt et al., 2003;

Girault et al., 2013a). Low microphytoplankton abundances (< 5 cell.cm$^{-3}$) in a coastal station of the NW Mediterranean Sea, even during the spring bloom (Gomez and Gorsky, 2003) and low abundances, 4 ± 5 and 3.6 ± 7 cell.cm$^{-3}$ , reported by Dugenne (2017) in the NW Mediterranean Sea, suggest that microphytoplankton are never dominant in this oligotrophic area and are rather dominated by diatoms and dinoflagellates (Ferrier-Pagès and Rassoulzadegan, 1994; Gomez and Gorsky, 2003; Marty et al., 2008).

**4.3.2 Horizontal and vertical distribution of the phytoplankton community structure**

A clear distinct tridimensional distribution of phytoplankton abundances was observed between the cold core and warm boundary waters. Despite apparent constant oligotrophy of the surface waters (Sect. 3.1.), high variations in phytoplankton assemblage structuration were evidenced in this study, consistently with previous studies led in similar oligotrophic areas (Maranon et al., 2003; Girault et al., 2013b).

Chl-*a* concentration, commonly used as a proxy for phytoplankton biomass was higher in the cold core waters compared to the warm boundaries. The cold core richness was sustained by higher *Prochlorococcus*, picoeukaryotes and nanoeukaryotes abundances (Fig. 5 and 6, Table 1). By contrast, high abundances of *Synechococcus* characterized the warm boundaries. The contrasted surface distribution between *Prochlorococcus* and *Synechococcus* populations is clearly visible on Fig. 6. As displayed by their vertical distribution (Fig. 8),

*Prochlorococcus* and picoeukaryotes higher abundances in the cold core waters resulted from upwelled nutrient rich waters. Maximal abundances above 80,000 and 4,000 cell.cm$^{-3}$ were recorded for *Prochlorococcus* and picoeukaryotes, respectively, at the DCM depth, where nitrates were not limiting but irradiance decreased (10-30% of surface PAR only). By contrast, *Synechococcus* presented low abundances at the DCM (< 5,000 cell.cm$^{-3}$, Fig. S2 in supplements) but maximal abundances (~ 30,000 cell.cm$^{-3}$) within the warm boundary mixed layer

(Fig. 8). *Prochlorococcus* and *Synechococcus* have demonstrated to occupy different light niches over the water



column (Agustí, 2004). *Synechococcus* are particularly adapted to depleted nitrate and phosphate conditions (Moutin et al., 2002; Michelou et al., 2011) and are severely light adapted due to less efficient accessory pigments (Moore et al., 1995). To acquire the necessary energy to grow, they have developed efficient ways to cope with light and UV stress, conversely to *Prochlorococcus* (Mella-Flores et al., 2012) which are able to grow

deeper in the euphotic zone (Olson et al., 1990a). Marty et al. (2008) reported similar vertical distribution patterns at the DYFAMED station in the central Ligurian subbasin under late summer/early fall conditions and such vertical distribution of the picophytoplankton has been described and explained in various other oligotrophic environments (Olson et al., 1990a; Campbell et al., 1997; Partensky et al., 1999; DuRand et al., 2001; Girault et al., 2013b). As a matter of fact, we have reported similar *Prochlorococcus* and *Synechococcus*

abundances ranging between 15,000 and 50,000 cells.mm$^{-3}$, although one or two order of magnitude between *Prochlorococcus* and *Synechococcus* abundances have been generally observed in strong to ultra-oligotrophic areas.

### 4.3.3 Contribution to total red fluorescence and C biomass

The FLR and C biomass contribution of Prochlorococcus, Synechococcus, picoeukaryotes and

nanoeukaryotes present opposite patterns than the one in abundances previously described between the cold core and warm boundary waters. Nanoeukaryote were the main contributors (>50%) in terms of pigment content (defined by FLR) and biomass. Marty et al. (2008) reported a 10% relatively constant contribution of C biomass for microphytoplankton in the same area during late summer/early fall based on pigment data analysis. Abundances of *Prochlorococcus* and *Synechococcus* throughout cold core and warm boundaries surface waters

were in the same order of magnitude than in their study ($10^5$ cell.cm$^{-3}$), but FLR and biomass contribution of *Prochlorococcus* were 5 to 10 times lower. When this contribution is integrated over the euphotic layer, studies led in similar oligotrophic environment indicated a larger contribution of *Prochlorococcus* to Chl-*a* and/or biomass compared to *Synechococcus* at this time of the year (Olson et al., 1990a; DuRand et al., 2001; Marty et al., 2008). In our study, as only surface data were considered, excluding the DCM phytoplankton assemblage, it

may explain the higher contribution of *Synechococcus* compared to *Prochlorococcus*.

### 4.3.4 Biology as a tracer at a fine-scale of water masses

*Synechococcus* relative contribution to total FLR as defined by AFCM tends to overestimate their importance compared to their contribution calculated from their cellular C quota. As *Synechococcus* are characterized by particular photosynthetic pigment compositions (phycoerythrin and phycocyanin, Olson et al.,



1990b), the relative higher FLR contribution could be explained by the phycocyanin red fluorescence emission into the red fluorescence channel. Indeed, few measurements were not considered due to the abnormal high FLR observed by AFCM for *Synechococcus* (Fig. 7a) causing a sudden increase in $FLR_{Total}$ (Fig. 7a) while no shift in red fluorescence was evidenced by the TSG in these type 2 waters (Sect. 4.1). A possible explanation for this

discrepancy, is that the TSG fluorometer measured red fluorescence emission > 685 nm, while the automated cytometer measured red fluorescence emission > 652 nm. *Synechococcus* photosynthetic pigment composition may vary depending on the strains or their growing condition (Olson et al., 1990b) and may also contain both phycoerythrin (PE) and phycocyanin (PC). PE emission maximum is located around 575 nm, while PC maximum fluorescence is around 650 nm. PC red fuorescence is therefore collected more efficiently by the

AFCM than the fluorometer and might explain the higher FLR collected by the AFCM. As some samples were also analysed on a FACSCalibur equipped with a 633 nm laser beam, it was possible to measure the red fluorescence induced by PC, and thus calculate the ratio PC/PE. It occurred that the *Synechococcus* population observed in type 2 waters (stations 6 and 7) had a higher PC:PE ratio (about 0.33, data not shown) compared to other stations (< 0.27, data not shown). The ratio PC:PE varies as a response to photoacclimatation, as well as to

chromatic adaptation (Dubinsky and Stamber, 2009; Stamber, 2013).

These *Synechococcus* populations were retrieved in the northern corners of our study area (Fig. 7c), characterized by warmer SST (>18.5°C) and lower SSS values (<38.24) than the rest of warm boundary waters. Fluorescence recorded in type 2 waters along the ship track by the TSG was not significantly different than in type 1 waters (Fig. 11b). Surface silicate concentrations in type 2 waters were the lowest observed (Fig. 11d). As

mentioned above, only few phytoplankton species requiring silicate (i.e. diatoms) were observed in the Ligurian subbasin at this time of the year, meaning that the silicate concentration values observed were unlikely induced by phytoplankton silicate consumption. Besides their apparent different physical properties, type 1 and 2 waters remained relatively close in terms of TSG fluorescence and phytoplankton abundances (Figure 11).

Figures 11i-l showed the mean cell $FLR_m$ of the *Prochlorococcus*, *Synechococcus*, picoeukaryotes and

nanoeukaryotes. The observed increase of *Prochlorococcus*, *Synechococcus* and picoeukaryotes cell $FLR_m$ in type 2 waters (Fig. 11i-k) might result from photoacclimation to depth by increasing their cell size and Chl-*a* per cell content (Olson et al., 1990b; Campbell et al. 1997; DuRand et al. 2001; Dubinsky and Stambler, 2009; Stambler, 2013), meaning that these surface waters were previously located in deeper low-light waters. However, type 2 warm boundary waters were characterized by the highest SST recorded during the campaign,

which implies that these waters were unlikely upwelled. Moreover, deep *Prochlorococcus* and *Synechococcus* cells located below the thermocline at the DCM were characterized by a ~5-fold higher FLR compared to surface



cells (Fig. S4). Vertical *Synechococcus* fluorescence recorded by benchtop flow cytometry at station 6 and 7 (type 2 warm boundary waters) were characterized by the highest values, down to 10 m depth, but remain still below the highest fluorescence recorded in the DCM, which reject the hypothesis of upwelled low light photoacclimated populations. Both cytometers working on fresh (AFCM) or fixed (benchtop cytometry) samples

observed higher *Synechococcus* FLR at station 6 and 7 in type 2 waters, and the benchtop flow cytometer highlighted their higher PC/PE ratios. The phytoplankton community in surface warm boundary waters 2 might then be considered as a distinct phytoplankton population, which grew in a different environment than warm boundary waters 1. These biological observations (fluorescence) made at the single cell level combined with the physical properties of surface type 2 warm boundary waters suggest that these surface waters have a distinct

origin and history than warm boundary waters 1.

Combining physical SST/SSS diagram (Fig. 7d) in which type 1 and 2 warm waters are not significantly distinguishable, with Ligurian subbasin surface circulation patterns (Introduction) and FLR anomalies (Fig. 7a), allow us to hypothesize that warm boundary waters 2 could then correspond to a patch of surface Thyrrhenian Sea brought by the Eastern Corsica Current trapped in MAW waters from the Western

Corsican Current. Although both warm boundary waters reflected similar biogeochemical growing conditions (depleted nitrate and phosphate surface waters) and phytoplankton group abundances, the distinct optical properties of phytoplankton groups recorded by flow cytometry combined to high-resolution observations could be the witness of a different (bio)geographical water mass origin.

**4.4 Flow cytometry and productivity estimates**

The application of a matrix growth population model based on high-frequency AFCM measurements in warm boundary surface waters provides estimates of daily production (division rate) and loss rate for *Prochlorococcus* and *Synechococcus* populations. The low in-situ growth rate obtained for *Prochlorococcus* ($\mu_{size}$=0.21 $d^{-1}$) and the higher growth rate ($\mu_{size}$=0.72 $d^{-1}$) got for *Synechococcus* corroborate their surface distribution pattern. The combination of surface growth rate and population's vertical distribution suggest that

*Prochlorococcus* growth was limited in warm boundary surface waters by more intense light conditions, whereas *Synechococcus* cells were more particularly adapted. *Synechococcus* growth rate was larger than one division per day (>0.69 $d^{-1}$). As expected for an asynchronous population, *Synechococcus* growth rate estimate from differences in minimal and maximal values of biovolume ($\mu_{ratio}$= 0.49) was smaller than the one retrieved from the size distribution variations $\mu_{size}$. For *Prochlorococcus*, both growth rates were characterized by low values.

Low size variations, close to the limits of detection of the flow-cytometer, might cause eventual bias in $\mu_{ratio}$



calculation. It could explain that $\mu_{ratio}$ (0.28 d$^{-1}$) was slightly higher than $\mu_{size}$. *Synechococcus* growth rate was consistent with values of 0.48-0.96 d$^{-1}$ reported by Ferrier-Pages and Rassoulzadegan (1994) and with the value of 0.6 d$^{-1}$ reported by Agawin et al. (1998) both measured at the same period in surface waters of coastal stations of NW Mediterranean Sea. *Prochlorococcus* growth rate was in the same range as the growth rate values

(between 0.1 and 0.4 d$^{-1}$) reported by Goericke et al. (1993) during summer and winter in surface waters of the Sargasso Sea. Vaulot et al. (1995) and Liu et al. (1997) measured *Prochlorococcus* growth rates of 0.5-0.7 d$^{-1}$ and 0.45-0.60 d$^{-1}$, respectively, in oligotrophic surface waters of the equatorial and subtropical Pacific, with abundances ranging from 50,000 to 200,000 cell.cm$^{-3}$. Riballet et al. (2015) found a linear relationship between SST and growth rate in October in the subtropical Pacific, with a growth rate value of ~0.4 d$^{-1}$ at 18°C. Vaulot et

al. (1995) reported maximal of growth rate values at 30 m depth, where *Prochlorococcus* abundances were the highest. Moore et al. (1995) noticed that LL *Prochlorococcus* strain growth could be limited by high light intensity and grew faster at lower light levels, whereas HL strain was photoinhibited only at the highest growth irradiance tested. The small growth rate of 0.21 d$^{-1}$ suggested that the surface layer is not the optimal environment for the growth of the *Prochlorococcus* observed. Thus, these *Prochlorococcus* could belong to the

LL *Prochlorococcus* strain. Maximal growth rates of the *Prochlorococcus* might be observed at the DCM, where maximal abundances were indeed observed.

*Prochlorococcus* loss rate (0.30 d$^{-1}$) was higher than its growth rate during our study, suggesting that loss processes in these surface waters tended to control the *Prochlorococcus* population abundance, resulting in a decrease in abundance. In the same time, *Synechococcus* loss rate was slightly lower (0.68 d$^{-1}$) than its growth

rate. Calculated loss rates include both biological factors (predation, viral lysis) and physical factors (removal or addition of cells through sedimentation, or physical transport). Our loss and growth rate estimates were relatively similar for both *Prochlorococcus* and *Synechoccocus* population. Similar observations were made by Hunter-Cevera et al. (2014) throughout a year on natural *Synechococcus* populations, using a similar approach. Riballet et al. (2015) reported a synchronization of *Prochlorococcus* cell production and mortality with the day/night

cycle in the subtropical Pacific gyre, which likely enforces ecosystem stability in oligotrophic ecosystems. In these ecosystems with limited submesoscale instabilities, picocyanobactoria abundances are relatively constant (Partensky et al., 1999a), as well as biogeochemical characteristics, on one to few days. The apparent equilibrium of cell abundances of these systems suggests that growth and loss processes are tightly coupled, which helps to stabilize open ocean ecosystems (Partensky et al., 1999a; Riballet et al., 2015).

Despite similar range of abundances of both picocyanobacteria (10,000-20,000 cells.cm$^{-3}$) the apparent productions NPP$_{size}$ and NPP$_{cell}$ of *Prochlorococcus* and *Synechococcus* (Table 3) indicate that *Synechococcus*



contribution to net C uptake was 20-25 times higher than *Prochlorococcus* in surface warm boundary waters. Following the growth rate difference previously described, it may reflect that environmental conditions in these surface waters favor the production of *Synechococcus* cells. Our NPP estimates for *Synechoccocus* (2.68 mgC.m$^{-3}$.d$^{-1}$, Table 3) were consistent with gross production between 1 and 4 mgC.m$^{-3}$.d$^{-1}$ reported by Agawin et al.

(1998) in the NW Mediterranean Sea at the same period. Marty et al. (2008) estimates of primary production in the Ligurian subbasin in summer/fall yielded to values comprised between 8 and 16 mgC.m$^{-3}$.d$^{-1}$ in surface waters. According to these estimates, apparent production of *Prochlorococcus* and *Synechoccocus* accounted to 0.5-1% and 17-33% of primary production, respectively, which is consistent with their relative contributions to i) total fluorescence of 2.5% and 33.3%,respectively, and ii) to C biomass of 4% and 22%, respectively, in surface

warm boundary waters mentioned in Sect. 3.4. Picocyanobacteria apparent net production rates obtained from different calculations (NPP$_{size}$ and NPP$_{cell}$, Eq. (8) and (9)) provide similar specific C uptake rates, meaning that the quantity of C assimilated during the photoperiod is strictly equivalent to the biomass of newly formed cells after mitosis. This result strengthens the characterization of oligotrophic ecosystems in which populations follow a daily dynamic at equilibrium.

However, our apparent production estimates for both *Prochlorococcus* and *Synechococcus* undergo several limitations. The successive conversions from FWS to biovolume and then to C contents remain a substantial source of uncertainty, although our cellular C quota are in agreement with the literature (Table 2). Recent advances in flow cytometry provide direct measurements of specific phytoplankton biomass on sorted populations (Graff et al., 2012). Growth rates do not account for size-specific removal processes (selective

grazing, sinking rates). A size selective grazing may alter *in-situ* growth rates by up to 20% of the estimation (Dugenne, 2017). To overcome this issue, Hunter-Cevera et al. (2014) performed dilution experiment to estimate the selective grazing rates. During the OSCAHR campaign, the study of the diel variation of cell size distribution was limited to the warm boundary surface waters based on the assumption that the picophytoplankton populations presented the same cellular properties across this hydrographical province. Tracking of coherent

time series in a particular zone based on an adaptive Lagrangian approach might be considered. That was the plan for OSCAHR but the bad weather conditions prevented it. The production estimates presented in this study rely on C conversions based on cell size, whereas many production estimates are still based on Chl-*a* to C conversion factors. Direct integration of growth rates in biogeochemical models (Cullen et al., 1993) and comparison to C-based productivity models (Westberry et al., 2008) should be envisaged for a better assessment

of the biogeochemical contribution of picocyanobacteria in oligotrophic ecosystems. Our estimates of specific growth rates and associated apparent production provide new insight into *Prochlorococcus* and *Synechoccocus*



population dynamics and will allow better understanding and quantifying of their respective biogeochemical and ecological contributions in oligotrophic ecosystems, where they play a major role.

**5 Conclusion**

The scientific objectives of the project OSCAHR (Observing Submesoscale Coupling At High
Resolution) were to characterize a submesoscale (fine-scale) dynamical structure and to study its influence on the distribution of biogenic elements and the structure and dynamics of the first trophic levels associated with it. The methodology included the use of novel platforms of observation for sampling the ocean surface layer at a high spatial and temporal frequency. In particular, a MVP (Moving Vessel Profiler) was deployed with CTD, Fluorescence and LOPC (Laser Optical Particle Counter) sensors. Furthermore, a new version of automated flow
cytometer optimized for small and dim cells was installed and tested for real-time, high-throughput sampling of phytoplankton functional groups, from micro-phytoplankton down to picocyanobacteria (including *Prochlorococcus*). Two sources of seawater have been used in OSCAHR: along with the onboard surface water intake, a new pumping system was developed and tested in order to sample the upper water column to one meter resolution.

The cruise strategy utilised an adaptive approach based on both satellite and numerical modeling data to identify a dynamical feature of interest and track its evolution. We have demonstrated that subsurface cold waters reached the surface in the centre of a cyclonic recirculation into the Ligurian subbasin. These nutrient-rich upwelled waters induced an increase of Chl-*a* concentration, and associated primary production, in the centre of the structure, whereas surrounding warm and oligotrophic boundary waters remained less productive. The
phytoplankton community structure was dominated in terms of abundance by *Prochlorococcus*, *Synechococcus*, pico- and nano-eukaryotes, respectively. *Prochlorococcus* and *Synechococcus* abundances exhibited an opposite distribution throughout cold and warm surface waters, with dominance of *Prochlorococcus* in cold core waters and of *Synechococcus* in warm boundary waters. These shifts fitted perfectly with the short terms transitions when passing through one water type to another. The study of the fine-scale vertical distribution of
*Prochlorococcus* and *Synechococcus* showed that the dominance of *Prochlorococcus* vs. *Synechococcus* in cold core waters was closely linked to the upwelled subsurface waters. Coupling cell's optical properties and physical properties appears a valuable approach for characterizing the origin of distinct surface waters types.

The OSCAHR campaign perfectly encompasses the new opportunity offered by coupling fine-scale vertical and horizontal physical measurements, remote sensing, modeled data, *in-situ* AFCM and

biogeochemistry using an innovative adaptive sampling strategy, in order to deeply understand the fine-scale dynamics of the phytoplankton community structure. The unprecedented spatial and temporal resolution obtained thanks to last advances in AFCM deployment allowed us to clearly demonstrate the preponderant role of physical fine-scale processes on the phytoplankton community structure distribution. For the first time, using

this new model of Cytobuoy commercial AFCM, we were able to fully resolve *Prochlorococcus* and *Synechococcus* picocyanobacteria, the smallest photoautotrophs on earth, which play a major role in widespread ocean oligotrophic areas. Finally, single cell analysis of well-defined *Prochlorococcus* and *Synechococcus* functional groups associated to a size structure population matrix model provided some precious indications of the daily dynamics of these populations. Primary productivity estimates of these two major phytoplankton

species obtained by this model are essential to better understand the contribution of picocyanobacteria to biological productivity. This study encourages continuing and improving such strategy to biogeochemically quantify the contribution of such fine-scale structures in the global ocean. Finally, repeated surveys of the phytoplankton community structure using this kind of combined approach will allow a better assessing of the impact of climate change and anthropogenic forcings. This is particularly of importance in the Mediterranean

Sea, which is a biodiversity hotspot under intense pressure from anthropogenic impacts and already one of the most impacted seas in the world (Lejeusne et al., 2009).

**Acknowledgements**

We thank Anne Petrenko, Louise Rousselet, Alain De Verneil, and Christophe Yohia and Christelle Pinazo for modeling outputs. We also thank Nicole Garcia for Chl-*a* measurements. Special thanks go to the DT-INSU

people from la Seyne sur Mer, and in particular to Malika Oudia for her help in administration work and Céline Heyndrickx and Frédéric Le Moal for their technical support. Genavir and in particular J. Fenouil are acknowledge for providing and assisting us with the MVP. MVP and associated captors have been brought by IFREMER and LOPB with co-funding of the Centre Européen de Technologies Sous-Marine (CETSM – Contrat de Projet Etat Région 2007-2013 en PACA) and the French ANR FOCEA (Project ANR-09-CEXC-006-01 to

M. Zhou and F. Carlotti). We also thank J. Farrar (MIT) for his suggestions on the cruise strategy, Alain Lefèbvre (IFREMER) for the Pocket FerryBox, and Frédéric Partensky for his constructive discussion. The OSCAHR cruise has been supported by MIO "Axes Transverses" program, by FEDER fundings (PRECYM flow cytometry platform) and by the following projects: CHROME (PI M. Thyssen, funded by Excellence Initiative of Aix-Marseille University - A*MIDEX, a French "Investissements d'Avenir" programme), SeaQUEST (PI O.



Ross, funded by UE FP7-People), AMICO (PI C. Pinazo, funded by Copernicus – MEDDE French Ministery MDE), BIOSWOT (PI F. d'Ovidio, funded by CNES). We also thank the captain and crew of the Téthys II research vessel.

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





**Table 1. Mean and standard deviation values of SST, SSS, density, Chl-*a*, nitrate, nitrite, phosphate and silicate in cold core and warm boundaries 1 and 2 during the OSCAHR campaign. *Prochlorococcus*, *Synechococcus*, PicoEukaryotes (PicoE, PicoHighFLR, PicoHighFLO), NanoEukaryotes (NanoE, NanoFLO, NanoHighFLO) and MicroEukaryotes (MicroE, MicroHighFLO) abundances are expressed in number of cells (N) per cm$^{-3}$, and mean red**
5 **fluorescence of *Prochlorococcus*, *Synechococcus*, PicoE and NanoE are expressed in arbitrary units (a.u.) per cell for each hydrographical province.**

| | | | Cold Core | Warm Boundary 1 | Warm Boundary 2 |
|---|---|---|---|---|---|
| | Number of samplings | | 76 | 78 | 23 |
| | SST (°C) | | $16.3 \pm 0.3$ | $17.9 \pm 0.5$ | $18.8 \pm 0.1$ |
| | SSS | | $38.19 \pm 0.02$ | $38.26 \pm 0.02$ | $38.22 \pm 0.01$ |
| | Density | | $1028.1 \pm 0.1$ | $1027.8 \pm 0.1$ | $1027.6 \pm 0.0$ |
| | Chl-$a$ (µg.dm$^{-3}$) | | $0.17 \pm 0.04$ | $0.11 \pm 0.03$ | $0.12 \pm 0.01$ |
| | $NO_3^-$ (µmol.kg$^{-1}$) | | $< 0.05$ | $< 0.05$ | $<0.05$ |
| | $NO_2^-$ (µmol.kg$^{-1}$) | | $< 0.05$ | $< 0.05$ | $< 0.05$ |
| | $PO_4^{3-}$ (µmol.kg$^{-1}$) | | $< 0.05$ | $< 0.05$ | $< 0.05$ |
| | $Si(OH)_4$ (µmol.kg$^{-1}$) | | $1.31 \pm 0.05$ | $1.20 \pm 0.05$ | $1.13 \pm 0.04$ |
| *Prochlorococcus* | Abundance (N.cm$^{-3}$) | x $10^4$ | $3.5 \pm 0.8$ | $2.0 \pm 0.6$ | $2.4 \pm 0.2$ |
| | Mean Red Fluorescence (a.u. cell$^{-1}$) | x $10^1$ | $4.3 \pm 0.8$ | $3.7 \pm 0.6$ | $4.8 \pm 0.5$ |
| *Synechococcus* | Abundance (N.cm$^{-3}$) | x $10^4$ | $1.8 \pm 0.3$ | $2.5 \pm 0.3$ | $3.1 \pm 0.2$ |
| | Mean Red Fluorescence (a.u. cell$^{-1}$) | x $10^2$ | $5.0 \pm 0.4$ | $4.2 \pm 0.6$ | $5.9 \pm 1.0$ |
| PicoEukaryotes | Abundance (N.cm$^{-3}$) | x $10^3$ | $1.5 \pm 0.2$ | $1.1 \pm 0.1$ | $1.2 \pm 0.1$ |
| | Mean Red Fluorescence (a.u. cell$^{-1}$) | x $10^3$ | $3.5 \pm 0.3$ | $3.3 \pm 0.5$ | $4.5 \pm 0.6$ |
| NanoEukaryotes | Abundance (N.cm$^{-3}$) | x $10^2$ | $8.9 \pm 0.8$ | $7.8 \pm 1.3$ | $8.1 \pm 0.6$ |
| | Mean Red Fluorescence (a.u. cell$^{-1}$) | x $10^4$ | $2.0 \pm 0.2$ | $1.8 \pm 0.2$ | $2.0 \pm 0.2$ |
| MicroEukaryotes | Abundance (N.cm$^{-3}$) | x $10^1$ | $2.8 \pm 0.4$ | $2.7 \pm 0.3$ | $2.5 \pm 0.3$ |




**Table 2. Mean and standard deviation of forward scatter (FWS), equivalent spherical diameter (ESD) and biovolume of Prochlorococcus, Syenchococcus, PicoEukaryotes (PicoE) and NanoEukaryotes (NanoE) during the OSCAHR campaign. ESD were computed according to the power law relationship (log(Size)=0.309\*log(FWS)-1.853, n = 17, $r^2$ = 0.94) obtained with silica beads of known diameter. Biovolumes were calculated considering that the cells were** 5 **spherical. Biovolumes were converted into mean carbon cellular quota ($Q_{C, calc}$) according to the $Q_{C, calc}$=a.Biovolume$^b$ relationship using conversion factors a and b reported by (1) Menden-Deuer and Lessard (2000). Carbon cellular quota ($Q_{C, lit}$, lit for literature) from (2) Campbell et al. (1994), (3) Shalapyonok et al. (2001) and (4) Reifel et al. (2014) were reported for comparison.**

|  | *Prochlorococcus* | *Synechococcus* | *PicoEukaryotes* | *NanoEukaryotes* |
|---|---|---|---|---|
| FWS (a.u. cell$^{-1}$) | 48 ± 21 | 357 ± 335 | $1.0 \times 10^4$ ± $0.6 \times 10^4$ | $4.0 \times 10^4$ ± $1.7 \times 10^4$ |
| ESD (µm) | 0.5 ± 0.1 | 0.9 ± 0.2 | 2.6 ± 0.5 | 4.1 ± 0.5 |
| Biovolume (µm$^3$.cell$^{-1}$) | 0.07 ± 0.03 | 0.46 ± 0.38 | 10.5 ± 5.5 | 37.0 ± 14.7 |
| Conversion coefficients (a, b) | (0.26, 0.86)[1] | (0.26, 0.86)[1] | (0.26, 0.86)[1] | (0.433, 0.863)[1] |
| $Q_{C, calc.}$ (fg C cell$^{-1}$) | 25 | 109 | 1880 | 9000 |
| $Q_{C, lit.}$ (fg C cell$^{-1}$) | 53[2] | 100[3]-250[2] | 2108[2] | 9160[4] |



**Table 3.** *Prochlorococcus* and *Synechococcus* **daily growth rate estimate ($\mu_{ratio}$) computed as the median size ratio $\mu_{ratio} = ln(\bar{v}_{max}/\bar{v}_{min})$, intrinsic growth rate ($\mu_{size}$) and loss rate ($l$) obtained from Eq. 7. NPP$_{cell}$ and NPP$_{size}$ biomass production values obtained from Eq. 8 and 9, respectively.**

|  | *Prochlorococcus* | *Synechococcus* |
|---|---|---|
| $\mu_{ratio}$ (d$^{-1}$) | 0.28 | 0.49 |
| $\mu_{size}$ (d$^{-1}$) | 0.21 | 0.72 |
| $l$ (d$^{-1}$) | 0.30 | 0.68 |
| NPP$_{cell}$ (mg C. m$^{-3}$. d$^{-1}$) | 0.11 | 2.68 |
| NPP$_{size}$ (mg C. m$^{-3}$. d$^{-1}$) | 0.13 | 2.80 |




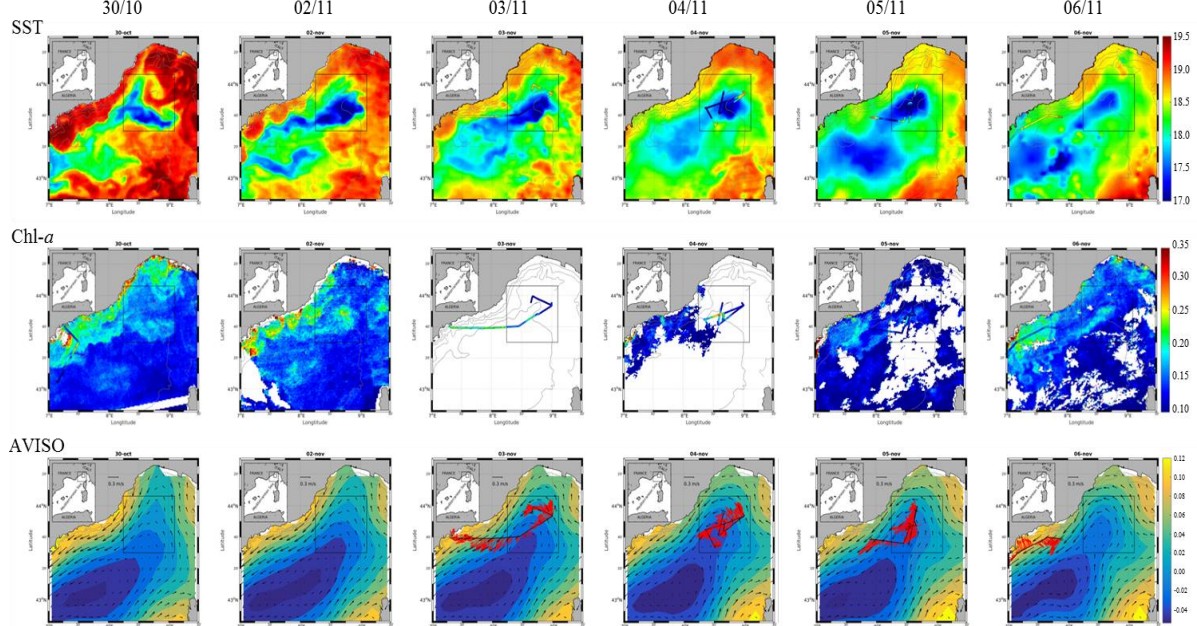

**Figure 1. Sea surface temperature (SST, in °C), Chl-a concentration (in µg.dm$^{-3}$) and AVISO altimetry (in cm) and derived currents intensity (m.s$^{-1}$) and direction in the Ligurian subbasin from the 30/10 to the 6/11. The black box represents the study area. From the 3/11 to the 6/11, SST and Chl-a continuous surface measurements were superimposed on the satellite products and ADCP currents were represented on the AVISO products.**



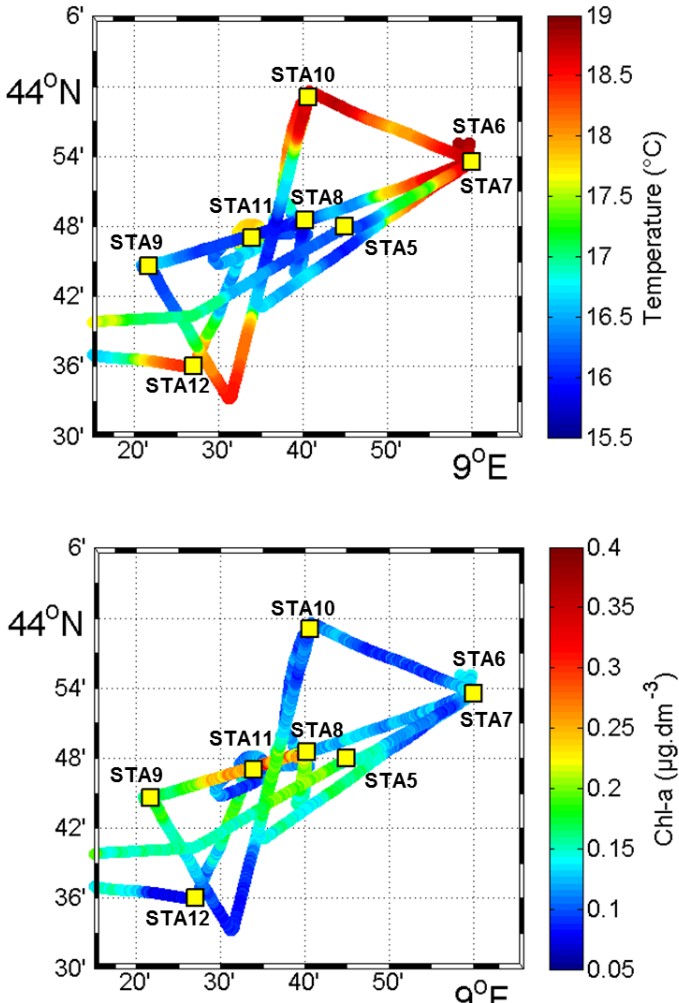

**Figure 2. Sea surface temperature (SST, in °C) and Chl-*a* concentrations (µg.dm⁻³) obtained from fluorescence continuous surface measurements from the 3/11 to the 6/11 during the OSCAHR campaign and fixed stations locations (STA5 to STA12). This study area correspond to the black box represented on Figure 1.**



**Figure 3. Continuous vertical profiles of salinity and temperature from the surface to 300 m depth between the points A and B from 00:00 to 6:00 (local time) on the 5/11. Associated SST (in °C), SSS and Chl-a concentration (in µg.dm⁻³) from continuous surface measurements and abundances (in cell.cm⁻³) of *Prochlorococcus*, *Synechococcus*, picoeukaryotes (PicoEuk) and nanoeukaryotes (NanoEuk).**



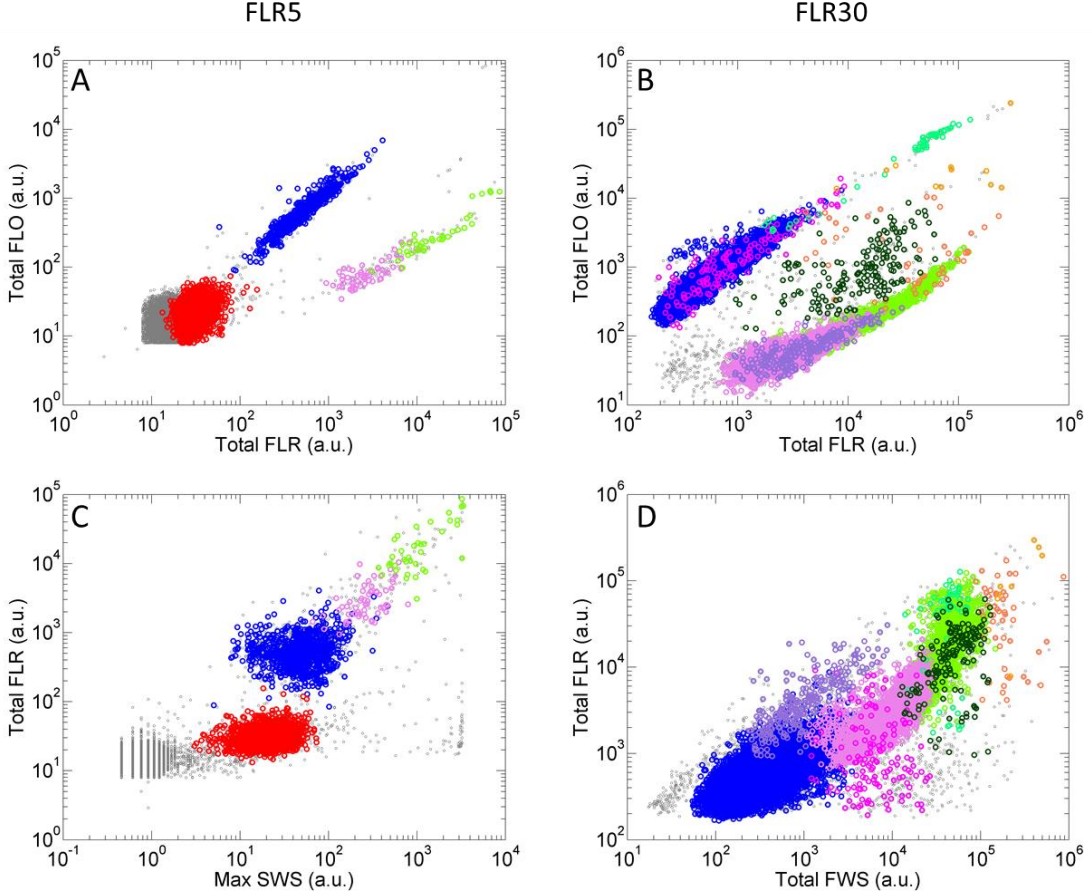

**Figure 4. Cytograms of samples analyzed with the CytoSense automated flow cytometer and phytoplankton groups optically resolved represented by different colors. Cytograms A and C were obtained with a red fluorescence (FLR) trigger level of 5 mV and cytograms B and D with a FLR trigger level of 30 mV. (A) Cytogram of total orange fluorescence (Total FLO (a.u.)) vs. Total FLR (a.u.). (B) Cytogram of Total FLO (a.u.) vs. Total FLR (a.u.). (C) Cytogram of Total FLR (a.u.)) vs. maximum sideward scatter (Max SWS (a.u.)). (D) Cytogram of Total FLR (a.u.)) vs. total forward scatter (Total FWS (a.u.)). *Prochlorococcus* cells are in red, *Synechococcus* in blue, the main picoeukaryote group (PicoE) in pink, picoeukaryotes with high FLO (PicoHighFLO) in fuchsia, picoeukaryotes with high FLR (PicoHighFLR) in mauve, the main nanoeukaryote group (NanoE) in green, nanoeukaryotes with intermediate FLO (NanoFLO) in dark green, nanoeukaryotes with high FLO (NanoHighFLO) in cyan, microeukaryotes (MicroE) in dark orange and microeukaryotes with high FLO (MicroHighFLO) in orange.**





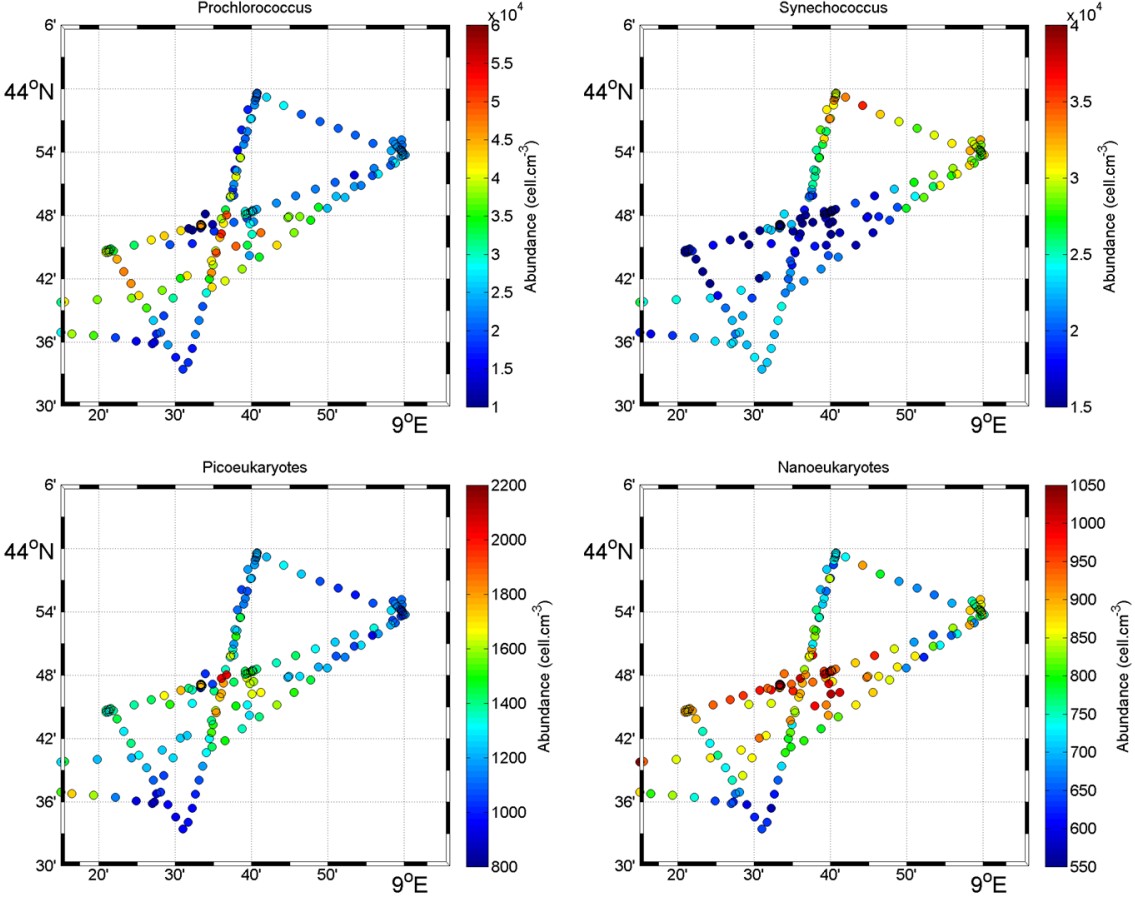

**Figure 5. Surface distribution of *Prochlorococcus, Synechococcus*, Picoeukaryotes (PicoE + PicoHighFLR + PicoHighFLO) and nanoeukaryotes (NanoE + NanoFLO + NanoHighFLO) abundances (in cells.cm⁻³).**





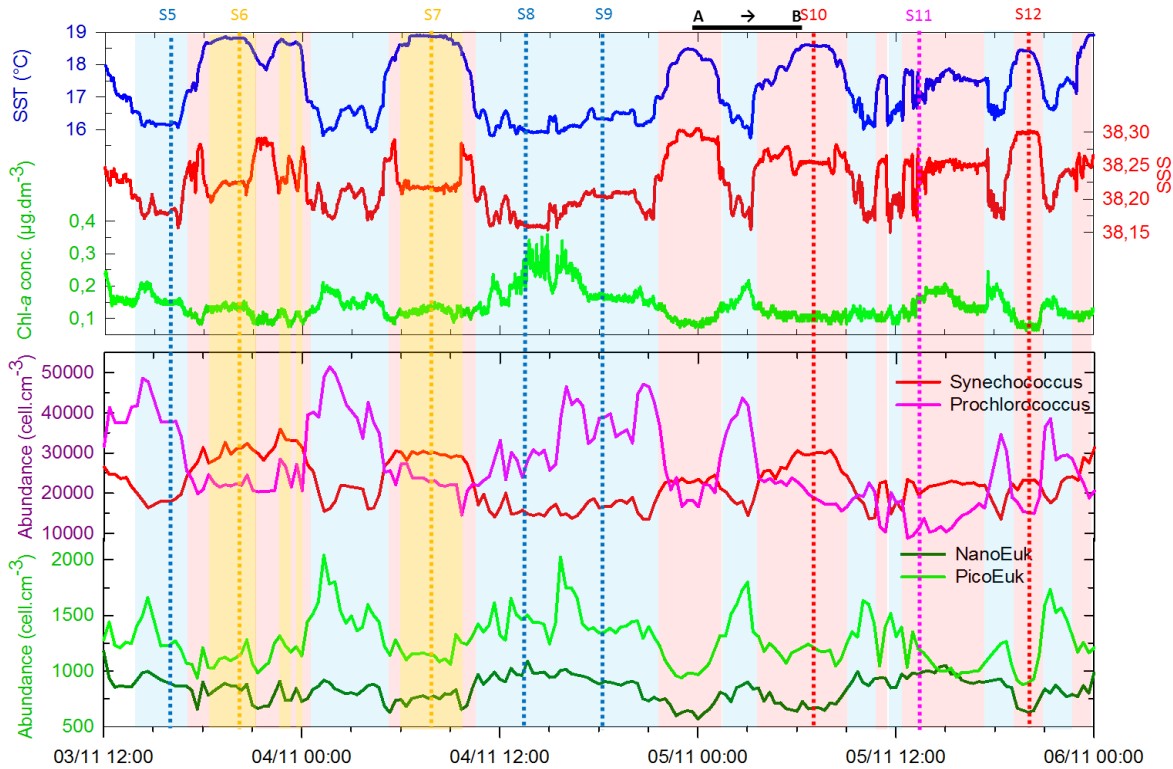

**Figure 6. Continuous measurements of SST (in °C), SSS and Chl-a concentrations (in µg.dm⁻³) of surface waters during the OSCAHR campaign from the 3/11 12:00 to the 6/11 00:00 (local time), with associated surface abundances (in cells.cm⁻³) of *Prochlorococcus*, *Synechococcus*, picoeukaryotes (PicoE + PicoHighFLO + PicoHighFLR) and nanoeukaryotes (NanoE + NanoFLO + NanoHighFLO). The background colorcode corresponds to cold core surface waters in blue, warm boundary waters of type 1 in red and warm boundary waters of type 2 in orange (more details in Sect. 4.2.). Vertical dashed lines represent sampling times of the 8 fixed stations (STA5 to STA12) performed during the campaign and colors correspond to the type of surface waters in which stations were performed. The purple color for STA11 exhibits that STA11 was performed in transitions surface waters between cold core and warm boundary 1 surface waters. Start and end of the MVP transect presented on Figure 3 are represented by a horizontal black line.**




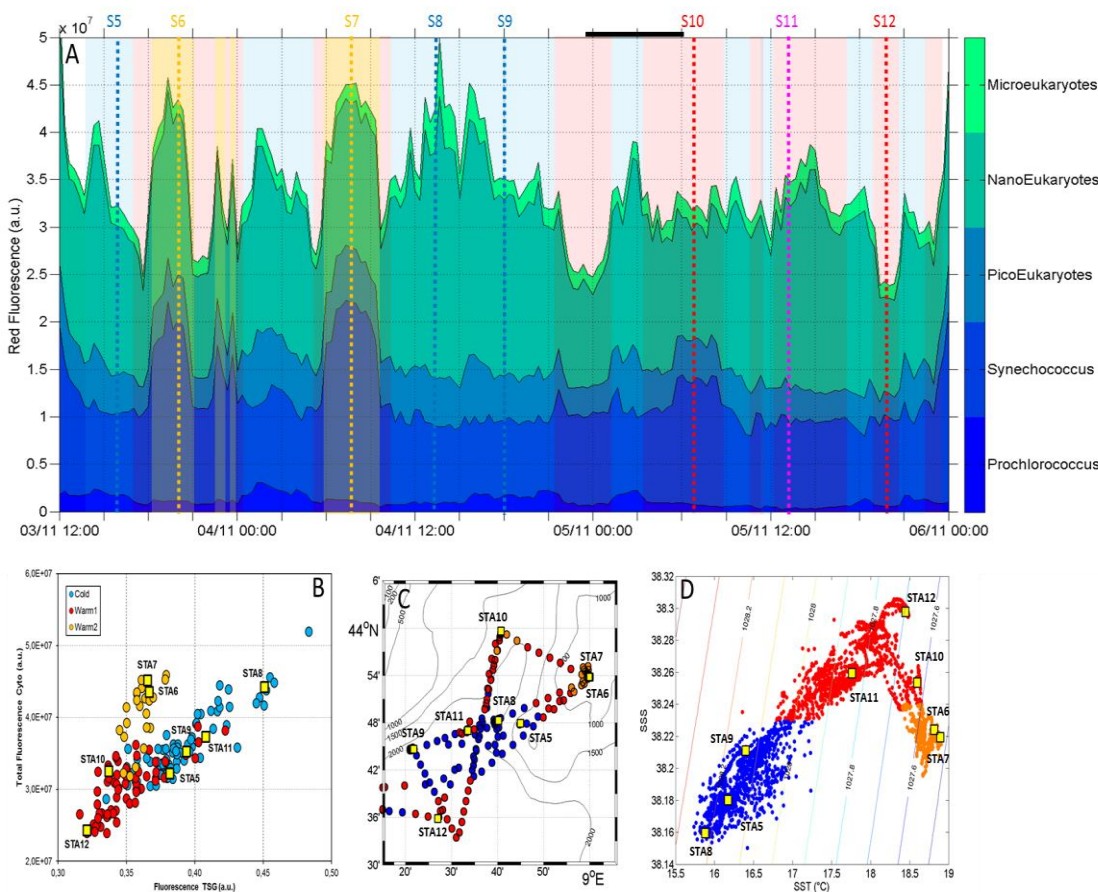

**Figure 7. (A)** Relative contribution $FLR_i = (FLR_{m,i} * Abundance_i)$ of *Prochlorococcus*, *Synechococcus*, picoeukaryotes (PicoE + PicoHighFLR + PicoHighFLO), nanoeukaryotes (NanoE + NanoFLO + NanoHighFLO) and microeukaryotes (MicroE + MicroHighFLO) to the integrated red fluorescence signal ($FLR_{Total} = \sum_i (FLR_{m,i} * Abundance_i)$)) from the 3/11 12:00 to the 6/11 00:00. Vertical dashed lines represent sampling times of the 8 fixed stations (STA5 to STA12) performed during the campaign and colors correspond to the type of surface waters in which stations were performed. **(B)** Fluorescence recorded with the FLRTotal (in a.u.) vs. TSG (in a.u.) recorded by the automated flow cytometer. Blue, red and orange dots correspond to sampling performed in cold core, warm boundary 1 and boundary 2 surface waters. **(C)** Sampling positions of automated flow cytometry surface measurements. Blue, red and orange dots correspond to sampling performed in cold core, warm boundary 1 and boundary 2 surface waters. **(D)** SSS vs. SST (in °C) plot from continuous TSG measurements with corresponding density isolines. The distinction between cold core, warm boundary 1 and 2 surface waters along the manuscript was done according to this plot




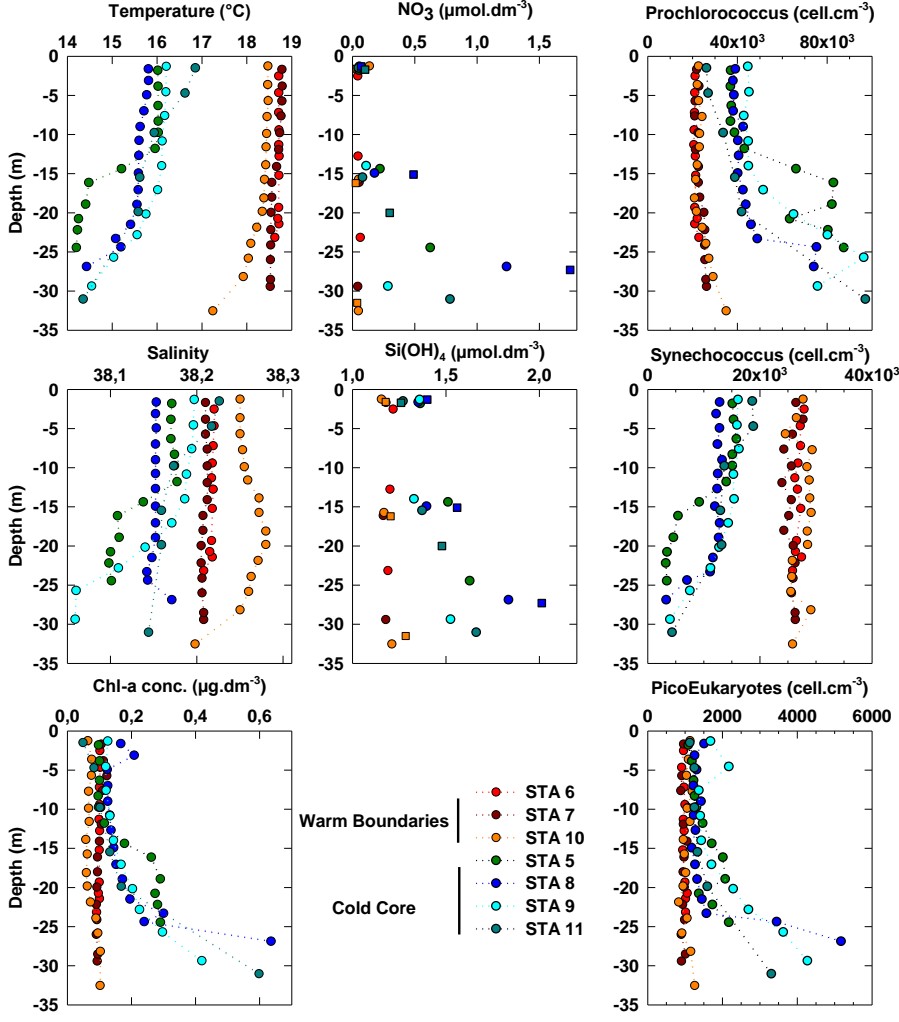

**Figure 8. Vertical profiles of temperature (in °C), salinity and Chl-a concentrations (in µg.dm$^{-3}$) obtained from the CTD fluorimeter after conversion at the depths where vertical high-resolution sampling were acquired for benchtop flow cytometry analysis using the PASTIS_HVR system. Abundances of *Prochlorococcus*, *Synechococcus* and picoeukaryotes (PicoE + PicoHighFLR + PicoHighFLO) groups are expressed in cells.cm$^{-3}$. Nutrients were sampled at a different resolution using both the PASTIS_HVR system (circles) and the CTD-rosette (squares). Stations performed in cold core surface waters are represented by blue-green colors and those performed in warm boundary surface waters by red-orange colors.**

500



**Figure 9. Observed (Obs.) and predicted (Pred.) hourly normalized cell size distributions (in μm³) of *Prochlorococcus* and *Synechococcus* from the 5/11 00:00 to the 6/11 00:00 (local time). White dots indicate the median size of the populations.**



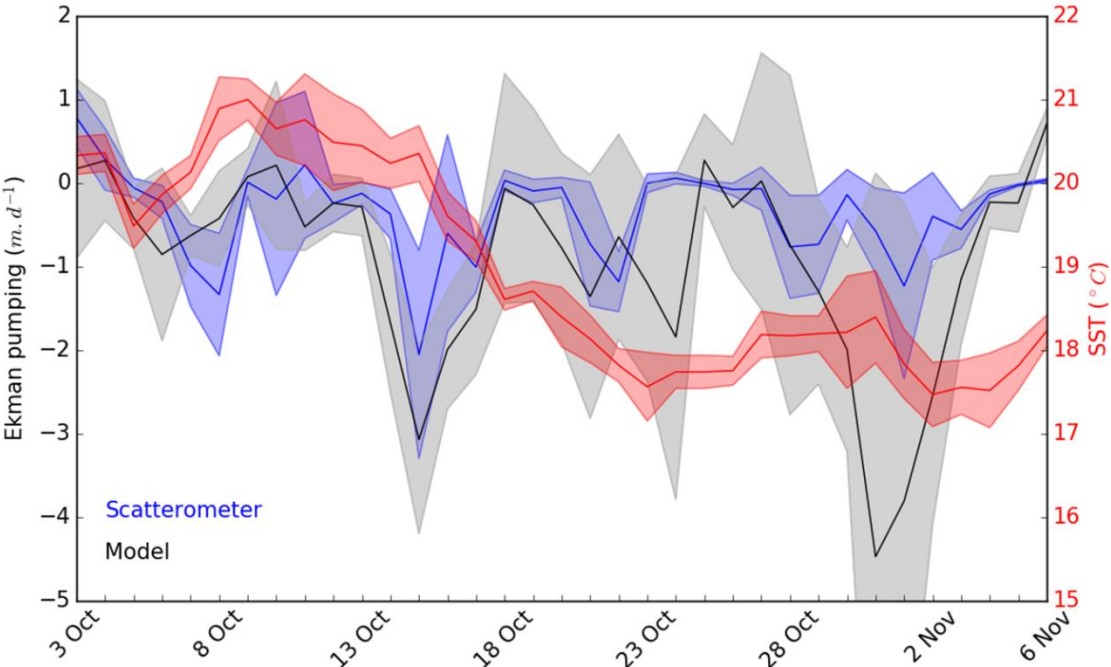

**Figure 10.** Ekman pumping vertical velocities (in m.d⁻¹) computed from scatterometer (in blue) and atmospheric model (in black) wind speeds and mean SST (in red, in °C) in our study area from the 3/10 to the 6/11. Shade areas represent the standard deviation relative to each measurement. Negative Ekman pumping values represent upward vertical velocities.



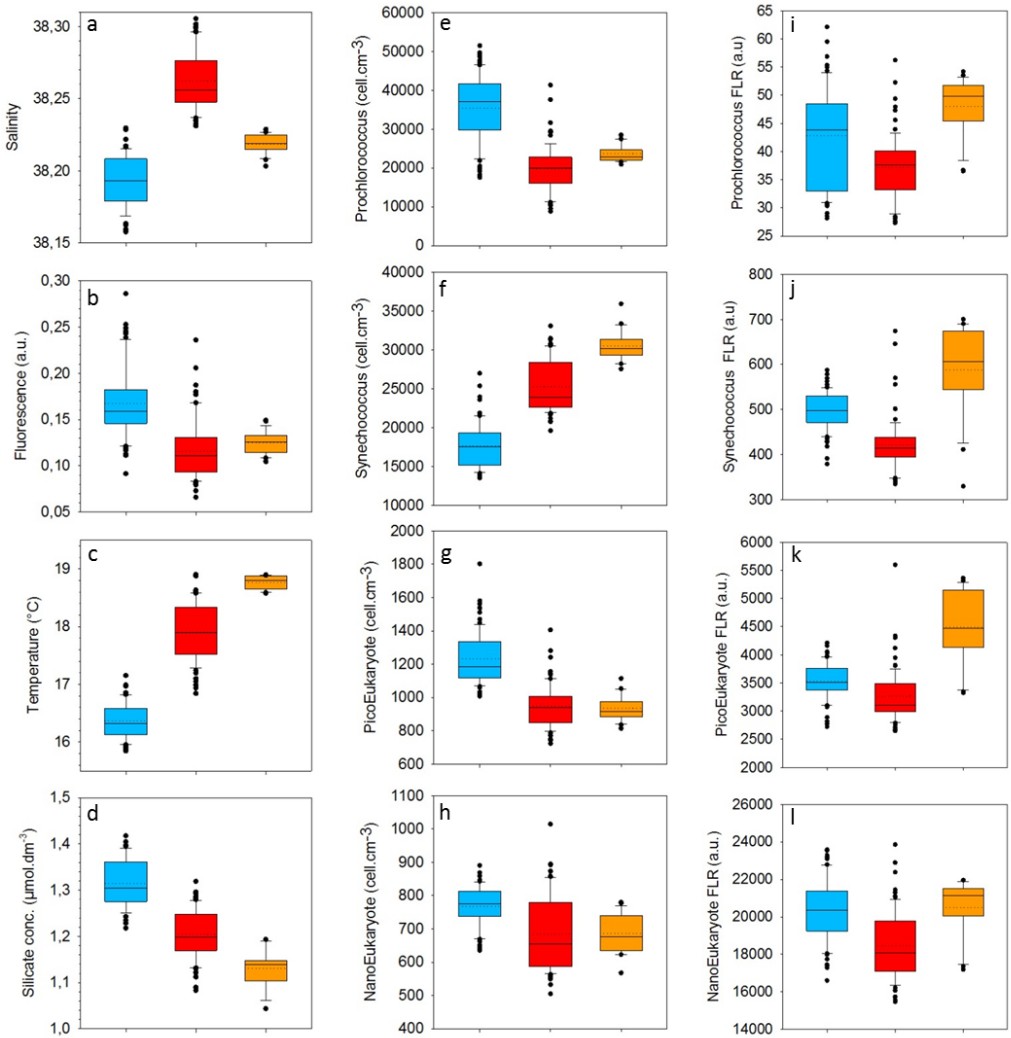

**Figure 11. Boxplots of SSS, fluorescence (in a.u.), SST (in °C) and silicate concentration (in μmol.dm$^{-3}$) in cold core (in blue), warm boundary 1 (in red) and 2 (in orange) surface waters. *Prochlorococcus*, *Synechococcus*, picoeukaryotes (PicoE) and nanoeukaryote (NanoE) abundances (in cell.cm$^{-3}$) and specific mean red fluorescence (FLR$_m$) in the same hydrographical provinces are also represented with boxplots. The boundary of the box closest to zero indicates the 25$^{th}$ percentile, the black line within the box marks the median, the dash line indicates the mean and the boundary of the box farthest from zero indicates the 75$^{th}$ percentile. Error bars above and below the box indicate the 90$^{th}$ and 10$^{th}$ percentiles and outlying points are represented. The number of observations on which are based these boxplots are reported in Table 1.**