# Peer review of "Coupling physics and biogeochemistry thanks to high resolution observations of the phytoplankton community structure in the North-Western Mediterranean Sea"

_Biogeosciences, 2017_

## Referee Comment (RC1) · Anonymous Referee #1 · 28 Sep 2017

This manuscript provides a valuable contribution to the study of the relationships between the fine scale distribution of physico-chemical variables and of flow cytometry-derived phytoplankton groups in open waters of the NW Mediterranean. The methodology is up to date and the measurements appear to have been carefully carried out. The conclusions are plausible, but it should be noted that there is more taxonomic richness in "phytoplankton community structure" than that measured in flow cytometric groups; it can be argued that some samples for microscopic examination (to name a classical technique) would have added interesting information to the work. The follow-

ing comments refer mainly to the "communication" aspect of the text, which is rather prolix and difficult to follow in several places. Methods Some parts of section 2.7 would benefit from more detailed and clearer explanations (e. g., lines 31 of page 8 to 3 of page 9). Some of the mathematical symbols used may not be obvious for a number of readers (e. g., eq. 5, eq. 9). Results Several parts of section 3.2 (Phytoplankton group definition) could be transferred to the Material and methods. (in particular, lines 1-20 of page 11). Lines 1-8 of page 13. There should be a previous explanation of what are warm boundary type 1 and type 2 waters (now in lines 34-39 of page 15). Section 3-5. Perhaps some of the details could be moved to material and Methods, so that the main findings would be easier to follow. Discussion Section 4.3. Part of the text is repetitive of methods or results and distracts from the main aim of the discussion. Please, try to streamline all the subsections. Other comments Page 1, line 29. "nanoeukaryotes". Page 2, line 5. "rise2. Page 5, line 9. The convenience of the phaeopigment "correction" is doubtful (e. g., Stich and Brinker 2995, Arch. Hydrobiol. 162 1 111–120). Line 39. SSS data every minute? Or what?? Page 6, lines 4-5. Rewrite the sentence. As it stands, it seems to say that 177 samples were collected every 20 minutes &e. g. "surface samples were collected every 20 minutes; in total, 177 were obtained" or similar). Line 14-15. "phytoplankton size wide range"??? or "a wide range of phytoplankton sizes"?. Line 36. Explain the meaning of "a.u." (arbitrary units?). Page 8, line 23. "cell removal processes". Page 10, lines 21-23. How exactly were these correlations carried out? Page 14, lines 8- 10. "although the sampling frequency spanned 20 min" ??? Explain better. Line 25. "derive growth rate". Page 15, line 24. "low salinity subsurface water". Lines 34-38. As mentioned before, this explanation should appear earlier. Page 16, line 8. "either limited"? Improve the sentence. Line 24. "resolve". Page 17, line 5. "ecotypes in surface waters". Line 17. "that the picoeukaryote". Page 17. Lines 17-21. Please, revise sentence carefully; concerning radiolarians and dinoflagellates, Not et al. (2009) state (page 4) that : "As the smallest eukaryotic organism known so far has a cell diameter of 0.8 mm [27], some of the 18S rDNA signatures observed in the 0.8 mm fraction might indeed derive from very small

eukaryotes (like the prasinophytes that appeared mostly in this small fraction, Table S4), but many sequences most likely derive from cell debris or extracellular DNA from larger cells. This is likely the case for radiolarians, dinoflagellates, and ciliates,groups known to contain relatively large nano- and microplanktonic cells, and for which sequences were prominent in the 0.8 mm fraction and nearly absent from the 0.8–3 m m fraction." (Thus, these groups were not part of the picoplankton). Not et al. (2009) also mention the importance of prasinophytes in the picoeukaryote fraction.

Page 17. Line 25: "cryptophycean taxa". Line 29: "Gephyrocapsa". Line 30: "Prymnesiophyceae". Page 18, line 5. Specify what is dominated by diatoms and dinoflagellates. Line 11 (and other parts of the text): "Marañón et al., 2003" as cited in the reference list (not "Maranon"). Line 19. "where nitrate was not limiting". Line 36. Italics for generic names.

---

## Referee Comment (RC2) · Anonymous Referee #2 · 3 Oct 2017

This paper presents a solid dataset collected during a cruise in the northwestern Med Sea, when a fine-scale physical structure (eddy) occurred. The authors describe the structure from different points of view and obtain a pretty exhaustive picture of its features, also tackling the potential functions exerted and providing rates estimates. The manuscript is interesting in its approach and provides useful information on biological functioning of eddies. In my opinion, the authors put too much emphasis on the technology used rather than on the results obtained, which could be eviscerated more. As a suggestion, they should make a stronger effort in building a global picture from their

data about how these eddies work and what contribution they bring to global ocean budgets. Specific comments follow: Abstract Line 15 – please define "fine-scale" Line 21. Synechococcus detection is not novel with these cytometers Line 23. It is not clear whose 1 m resolution belongs to. For a CTD is not much... Line 27 – replace "characterized with "and was marked"

Introduction In general, this section needs a reorganization to better harmonize the different topic presented Page 2 Line 17-18. This sentence is not clear, maybe you intend "Phytoplankton assemblages are highly..."? Line 26. Fine-scale variability of phytoplankton is known since more than a decade, e.g. work by Jim Mitchell, Laurent Seuront, just to name two. Page 3 Lines 7-8. I suggest to move "Eddy stirring...McGillicuddy) before "Mesoscale..." at line 3. Check spelling of McGillicuddy. Page 4 line 8. Should read "...depletion in surface waters..." Lines 12-22 I suggest to move this to page 3 line 16 and insert the info that you found a patch of cold water Lines 22-end. I would describe here the scientific aims of the cruise, not the list of ethodologies used Page 5 line 6. Delete "in relation with their environment" Results Page 13 line 16. Should define Case I waters or insert a reference Page 14 line 22. I suggest to modify as "A post-campaign validation against conventional flow cytometry showed a good fit of data (Student....Supplements)" Discussion Page 21 about the ecotypes of Prochlorochoccus. I am surprised that with the fine sampling resolution the two ecotypes are never seen together, as a bimodal distribution of red fluorescence. You may insert a short comment on this lack and possible explanations (have you observed them with conventional flow cytometry?)

---

## Author Comment (AC1) · 28 Nov 2017

Anonymous Referee #1 (AR1)

Referee Comment (RC): This manuscript provides a valuable contribution to the study of the relationships between the fine scale distribution of physico-chemical variables and of flow cytometry-derived phytoplankton groups in open waters of the NW Mediterranean. The methodology is up to date and the measurements appear to have been carefully carried out.

[Figure]

Authors Comment (AC): We really appreciate the positive and constructive comments addressed by anonymous referee #1. We would like to sincerely apologize for our relatively late responses regarding the reactive comments addressed by anonymous referee #1. This delayed response impeded a really interactive discussion between us, which is an important aspect of publishing in Biogeosciences. Your comments have allowed us to improve the overall quality of our manuscript. We have addressed all the comments relative to your recommendations below.

RC: The conclusions are plausible, but it should be noted that there is more taxonomic richness in "phytoplankton community structure" than that measured in flow cytometric groups; it can be argued that some samples for microscopic examination (to name a classical technique) would have added interesting information to the work.

AC: We acknowledge that there is more taxonomic richness in the phytoplankton community structure than that determined by the flow cytometric groups as optical properties measured by flow cytometry are ataxonomic (except for some specific genus such as Synechococcus and Prochlorococcus) and pictures taken in flow are adapted to microphytoplankton only. We will argue in the conclusion of the revised manuscript that optical microscope examination of samples might add interesting information but we will mention that according to the weak abundance of microphytoplankton (MicroE≈ 20 cells.cm-3 and MicroHighFLO < 5 cells.cm-3, with 10$\mu$m<MicroE ESD<20$\mu$m and MicroHighFLO ESD>20$\mu$m) and the small size of nanoeukaryote cells observed (ESD = 4.1±0.5 $\mu$m) a microscopic examination would also have been limited in resolution and quantification. Within our dataset, size classes between pico and nanophytoplankton (including pico and nanoeukaryotes cells and genus between Prochlorococcus and Synechococcus) present the main differences observed in the two contrasted areas with a high spatial resolution. Based on the literature, we briefly discuss the taxonomic richness in discussion section 4.3.1 from studies performed in the Mediterranean Sea in order to provide an overview of the main species that could have been found in the flow cytometric groups.

RC: The following comments refer mainly to the "communication" aspect of the text, which is rather prolix and difficult to follow in several places.

RC: Methods Some parts of section 2.7 would benefit from more detailed and clearer explanations (e. g., lines 31 of page 8 to 3 of page 9). Some of the mathematical symbols used may not be obvious for a number of readers (e. g., eq. 5, eq. 9).

AC: We acknowledge that some parts of Section 2.7 would benefit from more detailed and clearer explanation. In the revised manuscript we have addressed the requested modifications. We have also further described the meaning of the different mathematical symbols used in order to make this Section more accessible for some readers.

RC: Results Several parts of section 3.2 (Phytoplankton group definition) could be transferred to the Material and methods. (in particular, lines 1-20 of page 11).

AC: We agree that some parts of Section 3.2. could be transferred to the M&M Section. In the revised manuscript, lines 7-12 of page 14 have been included in the M&M Section. As this is the first deployment of this new model of AFCM, we considered that a technical description of the deployment and analysis of the AFCM could be included in the Result Section 3.2.

RC: Lines 1-8 of page 13. There should be a previous explanation of what are warm boundary type 1 and type 2 waters (now in lines 34-39 of page 15).

AC: We acknowledge that the explanation of what are warm boundary type 1 and 2 waters appears relatively late in the manuscript. The characterization of the warm boundary type 2 waters was supported by the study of the relative contribution to total red fluorescence which arrived later in the manuscript, and differentiating these 2 types of warm boundary waters only from the TS graph was kind of tricky. But we have introduced in the modified version these warm boundary type 1 and 2 waters from the 2nd paragraph of the Result Section 3.1.

RC: Section 3-5. Perhaps some of the details could be moved to material and Methods,

so that the main findings would be easier to follow.

AC: We agree that some details could be moved to M&M Section. We choose to move lines 15-17 of page 17 in Section 2.6.

RC: Discussion Section 4.3. Part of the text is repetitive of methods or results and distracts from the main aim of the discussion. Please, try to streamline all the subsections.

AC: We have reduced such repetitive parts in this section in the revised manuscript and we have streamlined all the subsections.

RC: Other comments

RC: Page 1, line 29. "nanoeukaryotes". AC: Done

RC: Page 2, line 5. "rise2. AC: Done.

RC: Page 5, line 9. The convenience of the phaeopigment "correction" is doubtful (e. g., Stich and Brinker 2995, Arch. Hydrobiol. 162 1 111–120). AC: We have modified this part of the Material and Methods Section and now we do not mention anymore the phaeopigment "correction" as it appears that the method used in our study is not exactly the one mentioned in our manuscript. We apologize for this misleading and thank you for your comment which allowed us to rectify this part of our manuscript.

RC: Line 39. SSS data every minute? Or what?? AC: Done

RC: Page 6, lines 4-5. Rewrite the sentence. As it stands, it seems to say that 177 samples were collected every 20 minutes &e. g. "surface samples were collected every 20 minutes; in total, 177 were obtained" or similar). AC: Thank you for your recommendation, we now mention that "surface samples were collected every 20 minutes. In total, up to 177 samples were obtained".

RC: Line 14-15. "phytoplankton size wide range"??? or "a wide range of phytoplankton sizes"?. AC: We meant "a wide range of phytoplankton sizes". Done

RC: Line 36. Explain the meaning of "a.u." (arbitrary units?). AC: Indeed, a.u. refers to arbitrary units.

RC: Page 8, line 23. "cell removal processes". AC: Done

RC: Page 10, lines 21-23. How exactly were these correlations carried out? AC: We apologize but we cannot find what this comment refers to due to some discrepancies between page and line numbers from your version of the manuscript and our version. Does your comment refer to the correlation between in-situ Chl-a and satellite values? Or between FLRtotal and Chl-a concentration? If it is about the in-situ vs. satellite Chl-a correlation, to compare in-situ observations with remote sensing products we extracted for each in-situ observation the closest one in time and space from the respective remote sensing product. We could add further details in the revised manuscript for this correlation. And if it is about the FLRtotal vs Chl-a correlation, we thought we have already provided enough details in our manuscript, but if needed we could eventually further describe the correlation.

RC: Page 14, lines 8- 10. "although the sampling frequency spanned 20 min" ??? Explain better. AC: We have modified this sentence in the revised manuscript by mentioning: "even if the sampling frequency spanned 20 min".

RC: Line 25. "derive growth rate". AC: Done

RC: Page 15, line 24. "low salinity subsurface water". AC: Done

RC: Lines 34-38. As mentioned before, this explanation should appear earlier. AC: The explanation of what are warm boundary type 1 and type 2 waters appears now earlier in the revised version of our manuscript, in the 2nd paragraph of Result Section 3.1.

RC: Page 16, line 8. "either limited"? Improve the sentence. AC: We now mention that: "This later was characterized by lower Chl-a values in the warm boundary, which was limited by both the nutrient availability and the amount of light availability for phytoplankton cells."

[Figure]

RC: Line 24. "resolve". AC: Done

RC: Page 17, line 5. "ecotypes in surface waters". AC: Done

RC: Line 17. "that the picoeukaryote". AC: Done

RC: Page 17. Lines 17-21. Please, revise sentence carefully; concerning radiolarians and dinoflagellates, Not et al. (2009) state (page 4) that : "As the smallest eukaryotic organism known so far has a cell diameter of 0.8 $\mu$m [27], some of the 18S rDNA signatures observed in the 0.8 $\mu$m fraction might indeed derive from very small eukaryotes (like the prasinophytes that appeared mostly in this small fraction, Table S4), but many sequences most likely derive from cell debris or extracellular DNA from larger cells. This is likely the case for radiolarians, dinoflagellates, and ciliates,groups known to contain relatively large nano- and microplanktonic cells, and for which sequences were prominent in the 0.8 $\mu$m fraction and nearly absent from the 0.8–3 $\mu$m fraction." (Thus, these groups were not part of the picoplankton). Not et al. (2009) also mention the importance of prasinophytes in the picoeukaryote fraction. AC: We thank you for this useful comment and apologize for our misinterpretation of this reference. We took notice of your recommendation and we have modified as requested this part of our discussion.

RC: Page 17. Line 25: "cryptophycean taxa". AC: Done

RC: Line 29: "Gephyrocapsa". AC: Done

RC: Line 30: "Prymnesiophyceae". AC: Done

RC: Page 18, line 5. Specify what is dominated by diatoms and dinoflagellates. AC: Microphytoplankton. Done

RC: Line 11 (and other parts of the text): "Marañón et al., 2003" as cited in the reference list (not "Maranon"). AC: Done

RC: Line 19. "where nitrate was not limiting". AC: Done

RC: Line 36. Italics for generic names. AC: Done 

---

## Author Comment (AC2) · 28 Nov 2017

Referee Comment (RC): This paper presents a solid dataset collected during a cruise in the northwestern Med Sea, when a fine-scale physical structure (eddy) occurred. The authors describe the structure from different points of view and obtain a pretty exhaustive picture of its features, also tackling the potential functions exerted and providing rates estimates. The manuscript is interesting in its approach and provides useful

information on biological functioning of eddies. In my opinion, the authors put too much emphasis on the technology used rather than on the results obtained, which could be eviscerated more. As a suggestion, they should make a stronger effort in building a global picture from their data about how these eddies work and what contribution they bring to global ocean budgets.

Authors Comment (AC): We do appreciate the positive and constructive comments addressed by anonymous referee #2. We would like to sincerely apologize for our relatively late responses regarding the reactive comments addressed by anonymous referee #2. This delayed response impeded a really interactive discussion between us, which is an important aspect of publishing in Biogeosciences. Your comments have allowed us to improve the overall quality of our manuscript. We have addressed all the comments relative to your recommendations below. As we used an innovative approach by deploying simultaneously several novel platforms of observation, we wanted to fully describe our methods. However, as you mention, we emphasized too much on the technology used, which impacts the highlighting of the main scientific findings of our study. Reviewers #1 addressed us several comments in order to enhance the consideration of the main findings and the main aims of our discussion. We hope that, by taking into consideration his/her comments, it could partially fulfill your suggestion. We think that the description of how the fine-scale structure works might be sufficiently characterized in our study. We agree that we should further insist and discuss on the contribution of such structure in the global ocean budgets, even if, as mentioned by Mahadevan (2016) it is still difficult to quantify how fine-scale processes affect the global state of the ocean. The main goal of our study was to present and test an original combination of new approaches to better observe and characterize the Ocean, in order to better understand how a fine scale structure works in order to apply them at a larger scale to finally quantify the contribution of these processes at a global scale.

RC: Specific comments follow:

RC: Abstract Line 15 – please define "fine-scale" AC: With the term "Fine-scale" we

refer to ocean dynamics features occurring at scales smaller than about 100km; consequently, the term includes i) a fraction of the mesoscale processes (e.g. large coherent eddies), with scales close to the first internal Rossby radius and ii) the submesoscale processes, with scales smaller than the first internal Rossby radius (e.g. intense vortices, fronts and filaments). This description of the "fine-scale" term is now included in the introduction of our revised manuscript. We added at line ??? a recent reference concerning the fine-scale dynamics in the studied area (Morrow et al, 2017) https://www.ocean-sci.net/13/13/2017/os-13-13-2017.pdf. In addition, the approximate size and duration of the observed fine-scale structure is now mentioned in the revised manuscript in the Results 3.1. Section. We choose to use this term since the studied structure has a complex dynamics and the meso or submeso scale terms would be too restrictive. We are indeed working on a deeper physical description of this structure, but that work is beyond the main purpose of this manuscript.

RC: Line 21. Synechococcus detection is not novel with these cytometers. AC: In the revised manuscript we have mentioned: 'For the first time with this optimized version of the AFCM, we were able to fully resolved Prochlorococcus picocyanobacteria, in addition to the easily distinguishable Synechococcus." Indeed, with the previous versions of the Cytosense instruments, Prochlorococcus were out of reach (too dim and too small) and only a part of the Synechococcus were properly detected. The original Cytosense technology was optimized for large particles (large phytoplankton and chains of cyanobacteria).

RC: Line 23. It is not clear whose 1 m resolution belongs to. For a CTD is not much. AC: We acknowledge that our description of this sampling system was not clear in the abstract. In the revised manuscript we have mentioned: 'A high-resolution vertical pumping system deployed during fixed stations allowed to sample water at a fine-resolution (below 1 m).'

RC: Line 27 – replace "characterized with "and was marked" AC: Done

RC: Introduction In general, this section needs a reorganization to better harmonize the different topic presented. AC: We agree that some parts of the introduction needed some reorganization and we have performed some substantial modifications in the revised manuscript. We have merged and fully reorganized paragraphs 3 and 4 relative to the Mediterranean Sea and moved them just before Section 4.1. as an introduction to our discussion and less insisted on the methodology used (2 sentences deleted). We hope that these modifications and the consideration of the subsequent recommendations will allow to better emphasize on the scientific aims of our study.

RC: Page 2 Line 17-18. This sentence is not clear, maybe you intend "Phytoplankton assemblages are highly"? AC: Thank you for this suggestion.

RC: Line 26. Fine-scale variability of phytoplankton is known since more than a decade, e.g. work by Jim Mitchell, Laurent Seuront, just to name two. AC: Thank you for these two references. We took notice of them and these studies have been mentioned in the revised manuscript. We have modified this sentence according to your recommendation by mentioning the main findings of these reference studies (and from others too) and we have argued that during the last decade numerous studies focused on this fine-scale variability and more particularly on the fine-scale variability of the phytoplankton community structure. Although patchiness and fractal distribution of phytoplankton were observed and described since more than a decade (i.e. Platt, 1972), in our study (and in the references mentioned) the phytoplankton community is described at a functional or an ecological trait. That's why we choose to consider these last decade references only.

RC: Page 3 Lines 7-8. I suggest to move "Eddy Stirring . . .McGillicuddy)" before "Mesoscale . . ." at line 3. AC: Done

RC: Check spelling of McGillicuddy. AC: Done

RC: Page 4 line 8. Should read " depletion in surface waters " AC: Done

RC: Lines 12-22 I suggest to move this to page 3 line 16 and insert the info that you found a patch of cold water AC: Done

RC: Lines 22-end. I would describe here the scientific aims of the cruise, not the list of methodologies used. AC: We have better presented in the modified version the scientific aim of the cruise but we think that it is also important to describe the innovative methods used to address these scientific aims as it is a major innovatory aspect of this study.

RC: Page 5 line 6. Delete "in relation with their environment" AC: Done

RC: Results Page 13 line 16. Should define Case I waters or insert a reference AC: We choose to insert the Morel et al. (2006) reference.

RC: Page 14 line 22. I suggest to modify as "A post-campaign validation against conventional flow cytometry showed a good fit of data (Student ... Supplements)" AC: Done

RC: Discussion Page 21 about the ecotypes of Prochlorochoccus. I am surprised that with the fine sampling resolution the two ecotypes are never seen together, as a bimodal distribution of red fluorescence. You may insert a short comment on this lack and possible explanations (have you observed them with conventional flow cytometry?)

AC: We thank you very much for this comment. We have now included a new figure in the supplement materials (Fig. S5), which presents the FLR distribution of Prochlorococcus obtained from samples analysed by conventional flow cytometry (in the laboratory) in the cold core (STA9) and warm boundary waters (STA5) over the first 35 m using the PASTIS high vertical resolution sampling system. We have also represented the FLR vertical distribution at STA11, the only station where we sampled the water column beyond the DCM. We never observed in surface waters the occurrence of Prochlorococcus population with significantly higher FLR (and/or SWS) values (Fig. 12 for AFCM and S4 for conventional flow cytometry) which might be representative of the

LL ecotype. AFCM measurements were only performed on surface seawater samples using the flow-through water supply. At the opposite, conventional flow cytometry analyses were performed on the first 35 m and revealed that we could be in the presence of both ecotype (HL and LL) together around the mixed layer depth in cold core waters (from 15-20 m depth). However, we did not observe any clear bimodal distribution of FLR (or SWS, data not shown) signals (Fig. S4 and new Fig. S5 in supplements). The DCM (i.e. 40 m depth), where the LL ecotype is supposed to be the main ecotype, was sampled only at one occasion, during the STA11 CTD-Rosette (Fig. S2, S4 and S5). Campbell and Vaulot (1993, Fig. 4) clearly show that a bimodal distribution of FLR intensities can be observe when 2 ecotypes are present together in "similar" proportion around the DCM. By "similar", we mean a sufficient abundance of both ecotypes, which make possible to clearly identify the bimodal distribution of FLR. Blanchot and Rodier (1996) also identify such a bimodal distribution in few locations. They clearly explained that in other location, ecotypes (sub-populations) co-occurrence cannot be observed from bimodality of the FLR distribution because both ecotypes were not abundant enough to be clearly seen. In these locations both ecotypes still existed, but their concentrations were very different and thus the two peaks could not be evidenced, the larger peak overpassing the smaller one. In the revised manuscript we mention Fig. S5 and strengthen our discussion about the two Prochlorococcus ecotypes distribution over the water column.

Campbell, L., & Vaulot, D. (1993). Photosynthetic picoplankton community structure in the subtropical North Pacific Ocean near Hawaii (station ALOHA). Deep Sea Research Part I: Oceanographic Research Papers, 40(10), 2043-2060. Blanchot, J., & Rodier, M. (1996). Picophytoplankton abundance and biomass in the western tropical Pacific Ocean during the 1992 El Niño year: results from flow cytometry. Deep Sea Research Part I: Oceanographic Research Papers, 43(6), 877-895.

[Figure]

Figure S5: FLR distribution of Prochlorococcus populations at STA7 (warm boundary waters, in red) and at STA9 (cold core waters, in blue), expressed in terms of cell density. Data comes from conventional flow-cytometry measurements performed from 30 m depth to the surface using the PASTIS pumping system to collect the water at various depths. The dotted lines represent the mean of the normal distribution for Prochlorococcus surface ecotype (HL – High-Light) and the dashed line represents the mean of the normal distribution for Prochlorococcus deep ecotype (LL – Low-Light). The same representations for the deep-cast STA11 – CTD-rosette also reflects the presence of at least 2 different Prochlorococcus populations discriminated from the distribution of their FLR values. Co-occurrence of both ecotypes can be observed at STA9 and STA11 but a clear distinction of the FLR distribution of each ecotype is not possible.

[Figure]

[Figure]

**Fig. 1.** Fig S5

---

## Author Response (AR1)

This manuscript provides a valuable contribution to the study of the relationships between the fine scale distribution of physico-chemical variables and of flow cytometry-derived phytoplankton groups in open waters of the NW Mediterranean. The methodology is up to date and the measurements appear to have been carefully carried out.

We really appreciate the positive and constructive comments addressed by anonymous referee #1. We would like to sincerely apologize for our relatively late responses regarding the reactive comments addressed by anonymous referee #1. This delayed response impeded a really interactive discussion between us, which is an important aspect of publishing in Biogeosciences. Your comments have allowed us to improve the overall quality of our manuscript. We have addressed all the comments relative to your recommendations below.

The conclusions are plausible, but it should be noted that there is more taxonomic richness in "phytoplankton community structure" than that measured in flow cytometric groups; it can be argued that some samples for microscopic examination (to name a classical technique) would have added interesting information to the work.

We acknowledge that there is more taxonomic richness in the phytoplankton community structure than that determined by the flow cytometric groups as optical properties measured by flow cytometry are ataxonomic (except for some specific genus such as Synechococcus and Prochlorococcus) and pictures taken in flow are adapted to microphytoplankton only. We will argue in the conclusion of the revised manuscript that optical microscope examination of samples might add interesting information but we will mention that according to the weak abundance of microphytoplankton (MicroE≈ 20 cells.cm$^{-3}$ and MicroHighFLO < 5 cells.cm$^{-3}$, with 10μm<MicroE ESD<20μm and MicroHighFLO ESD>20μm) and the small size of nanoeukaryote cells observed (ESD = 4.1±0.5 μm) a microscopic examination would also have been limited in resolution and quantification. Within our dataset, size classes between pico and nanophytoplankton (including pico and nanoeukaryotes cells and genus between Prochlorococcus and Synechococcus) present the main differences observed in the two contrasted areas with a high spatial resolution. Based on the literature, we briefly discuss the taxonomic richness in discussion section 4.3.1 from studies performed in the Mediterranean Sea in order to provide an overview of the main species that could have been found in the flow cytometric groups.

The following comments refer mainly to the "communication" aspect of the text, which is rather prolix and difficult to follow in several places.

Methods Some parts of section 2.7 would benefit from more detailed and clearer explanations (e. g., lines 31 of page 8 to 3 of page 9). Some of the mathematical symbols used may not be obvious for a number of readers (e. g., eq. 5, eq. 9).

We acknowledge that some parts of Section 2.7 would benefit from more detailed and clearer explanation. In the revised manuscript we have addressed the requested modifications. We have also further described the meaning of the different mathematical symbols used in order to make this Section more accessible for some readers.

Results Several parts of section 3.2 (Phytoplankton group definition) could be transferred to the Material and methods. (in particular, lines 1-20 of page 11).

We agree that some parts of Section 3.2. could be transferred to the M&M Section. In the revised manuscript, lines 7-12 of page 14 have been included in the M&M Section. As this is the first deployment of this new model of AFCM, we considered that a technical description of the deployment and analysis of the AFCM could be included in the Result Section 3.2.

Lines 1-8 of page 13. There should be a previous explanation of what are warm boundary type 1 and type 2 waters (now in lines 34-39 of page 15).

We acknowledge that the explanation of what are warm boundary type 1 and 2 waters appears relatively late in the manuscript. The characterization of the warm boundary type 2 waters was supported by the study of the relative contribution to total red fluorescence which arrived later in the manuscript, and differentiating these 2 types of warm boundary waters only from the TS graph was kind of tricky. But we have introduced in the modified version these warm boundary type 1 and 2 waters from the 2$^{nd}$ paragraph of the Result Section 3.1.

Section 3-5. Perhaps some of the details could be moved to material and Methods, so that the main findings would be easier to follow.

We agree that some details could be moved to M&M Section. We choose to move lines 15-17 of page 17 in Section 2.6.

Discussion Section 4.3. Part of the text is repetitive of methods or results and distracts from the main aim of the discussion. Please, try to streamline all the subsections.

We have reduced such repetitive parts in this section in the revised manuscript and we have streamlined all the subsections.

Other comments

Page 1, line 29. "nanoeukaryotes". Done

Page 2, line 5. "rise2. Done.

Page 5, line 9. The convenience of the phaeopigment "correction" is doubtful (e. g., Stich and Brinker 2995, Arch. Hydrobiol. 162 1 111–120). We have modified this part of the Material and Methods Section and now we do not mention anymore the phaeopigment "correction" as it appears that the method used in our study is not exactly the one mentioned in our manuscript. We apologize for this misleading and thank you for your comment which allowed us to rectify this part of our manuscript.

Line 39. SSS data every minute? Or what?? Done

Page 6, lines 4-5. Rewrite the sentence. As it stands, it seems to say that 177 samples were collected every 20 minutes &e. g. "surface samples were collected every 20 minutes; in total, 177 were obtained" or similar). Thank you for your recommendation, we now mention that "surface samples were collected every 20 minutes. In total, up to 177 samples were obtained".

Line 14-15. "phytoplankton size wide range"??? or "a wide range of phytoplankton sizes"?. We meant "a wide range of phytoplankton sizes" Done

Line 36. Explain the meaning of "a.u." (arbitrary units?). Indeed, a.u. refers to arbitrary units.

Page 8, line 23. "cell removal processes". Done

Page 10, lines 21-23. How exactly were these correlations carried out? We apologize but we cannot find what this comment refers to due to some discrepancies between page and line numbers from your version of the manuscript and our version. Does your comment refer to the correlation between in-situ Chl-a and satellite values? Or between FLRtotal and Chl-a concentration? If it is about the in-situ vs. satellite Chl-a correlation, to compare in-situ observations with remote sensing products we extracted for each in-situ observation the closest one in time and space from the respective remote sensing product. We could add further details in the revised manuscript for this correlation. And if it is about the FLRtotal vs Chl-a correlation, we thought we have already provided enough details in our manuscript, but if needed we could eventually further describe the correlation.

Page 14, lines 8- 10. "although the sampling frequency spanned 20 min" ??? Explain better. We have modified this sentence in the revised manuscript by mentioning: "even if the sampling frequency spanned 20 min".

Line 25. "derive growth rate". Done

Page 15, line 24. "low salinity subsurface water". Done

Lines 34-38. As mentioned before, this explanation should appear earlier. The explanation of what are warm boundary type 1 and type 2 waters appears now earlier in the revised version of our manuscript, in the 2nd paragraph of Result Section 3.1.

Page 16, line 8. "either limited"? Improve the sentence. We now mention that: "This later was characterized by lower Chl-a values in the warm boundary, which was limited by both the nutrient availability and the amount of light availability for phytoplankton cells."

Line 24. "resolve". Done

Page 17, line 5. "ecotypes in surface waters". Done

Line 17. "that the picoeukaryote". Done

Page 17. Lines 17-21. Please, revise sentence carefully; concerning radiolarians and dinoflagellates, Not et al. (2009) state (page 4) that : "As the smallest eukaryotic organism known so far has a cell diameter of 0.8 μm [27], some of the 18S rDNA signatures observed in the 0.8 μm fraction might indeed derive from very small eukaryotes (like the prasinophytes that appeared mostly in this small fraction, Table S4), but many sequences most likely derive from cell debris or extracellular DNA from larger cells. This is likely the case for radiolarians, dinoflagellates, and ciliates,groups known to contain relatively large nano- and microplanktonic cells, and for which sequences were prominent in the 0.8 μm fraction and nearly absent

from the 0.8–3 µm fraction." (Thus, these groups were not part of the picoplankton). Not et al. (2009) also mention the importance of prasinophytes in the picoeukaryote fraction.

We thank you for this useful comment and apologize for our misinterpretation of this reference. We took notice of your recommendation and we have modified as requested this part of our discussion.

Page 17. Line 25: "cryptophycean taxa". Done

Line 29: "Gephyrocapsa". Done

Line 30: "Prymnesiophyceae". Done

Page 18, line 5. Specify what is dominated by diatoms and dinoflagellates. Microphytoplankton. Done

Line 11 (and other parts of the text): "Marañón et al., 2003" as cited in the reference list (not "Maranon"). Done

Line 19. "where nitrate was not limiting". Done

Line 36. Italics for generic names. Done

Anonymous Referee #2

This paper presents a solid dataset collected during a cruise in the northwestern Med Sea, when a fine-scale physical structure (eddy) occurred. The authors describe the structure from different points of view and obtain a pretty exhaustive picture of its features, also tackling the potential functions exerted and providing rates estimates. The manuscript is interesting in its approach and provides useful information on biological functioning of eddies. In my opinion, the authors put too much emphasis on the technology used rather than on the results obtained, which could be eviscerated more. As a suggestion, they should make a stronger effort in building a global picture from their data about how these eddies work and what contribution they bring to global ocean budgets.

We do appreciate the positive and constructive comments addressed by anonymous referee #2. We would like to sincerely apologize for our relatively late responses regarding the reactive comments addressed by anonymous referee #2. This delayed response impeded a really interactive discussion between us, which is an important aspect of publishing in Biogeosciences. Your comments have allowed us to improve the overall quality of our manuscript. We have addressed all the comments relative to your recommendations below.
As we used an innovative approach by deploying simultaneously several novel platforms of observation, we wanted to fully describe our methods. However, as you mention, we emphasized too much on the technology used, which impacts the highlighting of the main scientific findings of our study. Reviewers #1 addressed us several comments in order to enhance the consideration of the main findings and the main aims of our discussion. We hope that, by taking into consideration his/her comments, it could partially fulfill your suggestion. We think that the description of how the fine-scale structure works might be sufficiently characterized in our study. We agree that we should further insist and discuss on the contribution of such structure in the global ocean budgets, even if, as mentioned by Mahadevan (2016) it is still difficult to quantify how fine-scale processes affect the global state of the ocean. The main goal of our study was to present and test an original combination of new approaches to better observe and characterize the Ocean, in order to better understand how a fine scale structure works in order to apply them at a larger scale to finally quantify the contribution of these processes at a global scale.

Specific comments follow:

Abstract Line 15 – please define "fine-scale"
With the term "Fine-scale" we refer to ocean dynamics features occurring at scales smaller than about 100km; consequently, the term includes i) a fraction of the *mesoscale* processes (e.g. large coherent eddies), with scales close to the first internal Rossby radius and ii) the submesoscale processes, with scales smaller than the first internal Rossby radius (e.g. intense vortices, fronts and filaments). This description of the "fine-scale" term is now included in the introduction of our revised manuscript. We choose to use this term since the studied structure has a complex dynamics and the meso or submeso scale terms would be too restrictive. We are indeed working on a deeper physical description of this structure, but that work is beyond the main purpose of this manuscript.

Line 21. Synechococcus detection is not novel with these cytometers. In the revised manuscript we have mentioned: 'For the first time with this optimized version of the AFCM, we were able to fully resolved Prochlorococcus picocyanobacteria, in addition to the easily distinguishable Synechococcus." Indeed, with the previous versions of the Cytosense instruments, Prochlorococcus were out of reach (too dim and too small) and only a part of the Synechococcus were properly detected. The original Cytosense technology was optimized for large particles (large phytoplankton and chains of cyanobacteria).

Line 23. It is not clear whose 1 m resolution belongs to. For a CTD is not much. We acknowledge that our description of this sampling system was not clear in the abstract. In the revised manuscript we have mentioned: 'A high-resolution vertical pumping system deployed during fixed stations allowed to sample water at a fine-resolution (below 1 m).'

Line 27 – replace "characterized with "and was marked" Done

Introduction In general, this section needs a reorganization to better harmonize the different topic presented. We agree that some parts of the introduction needed some reorganization and we have performed some substantial modifications in the revised manuscript. We have merged and fully reorganized paragraphs 3 and 4 relative to the Mediterranean Sea and moved them just before Section 4.1. as an introduction to our discussion and less insisted on the methodology used (2 sentences deleted). We hope that these modifications and the consideration of the subsequent recommendations will allow to better emphasize on the scientific aims of our study.

Page 2 Line 17-18. This sentence is not clear, maybe you intend "Phytoplankton assemblages are highly"? Thank you for this suggestion.

Line 26. Fine-scale variability of phytoplankton is known since more than a decade, e.g. work by Jim Mitchell, Laurent Seuront, just to name two. Thank you for these two references. We took notice of them and these studies have been mentioned in the revised manuscript. We have modified this sentence according to your recommendation by mentioning the main findings of these reference studies (and from others too) and we have argued that during the last decade numerous studies focused on this fine-scale variability and more particularly on the fine-scale variability of the phytoplankton community structure.
Although patchiness and fractal distribution of phytoplankton were observed and described since more than a decade (i.e. Platt, 1972), in our study (and in the references mentioned) the phytoplankton community is described at a functional or an ecological trait. That's why we choose to consider these last decade references only.

Page 3 Lines 7-8. I suggest to move "Eddy Stirring …McGillicuddy)" before "Mesoscale …" at line 3. Done

Check spelling of McGillicuddy. Done

Page 4 line 8. Should read " depletion in surface waters " Done

Lines 12-22 I suggest to move this to page 3 line 16 and insert the info that you found a patch of cold water Done

Lines 22-end. I would describe here the scientific aims of the cruise, not the list of methodologies used We have better presented in the modified version the scientific aim of the cruise but we think that it is also important to describe the innovative methods used to address these scientific aims as it is a major innovatory aspect of this study.

Page 5 line 6. Delete "in relation with their environment" Done

Results Page 13 line 16. Should define Case I waters or insert a reference We choose to insert the Morel et al. (2006) reference.

Page 14 line 22. I suggest to modify as "A post-campaign validation against conventional flow cytometry showed a good fit of data (Student ... Supplements)" Done

Discussion Page 21 about the ecotypes of Prochlorochoccus. I am surprised that with the fine sampling resolution the two ecotypes are never seen together, as a bimodal distribution of red fluorescence. You may insert a short comment on this lack and possible explanations (have you observed them with conventional flow cytometry?)

We thank you very much for this comment. We have now included a new figure in the supplement materials (Fig. S5), which presents the FLR distribution of Prochlorococcus obtained from samples analysed by conventional flow cytometry (in the laboratory) in the cold core (STA9) and warm boundary waters (STA5) over the first 35 m using the PASTIS high vertical resolution sampling system. We have also represented the FLR vertical distribution at STA11, the only station where we sampled the water column beyond the DCM. We never observed in surface waters the occurrence of Prochlorococcus population with significantly higher FLR (and/or SWS) values (Fig. 12 for AFCM and S4 for conventional flow cytometry) which might be representative of the LL ecotype. AFCM measurements were only performed on surface seawater samples using the flow-through water supply. At the opposite, conventional flow cytometry analyses were performed on the first 35 m and revealed that we could be in the presence of both ecotype (HL and LL) together around the mixed layer depth in cold core waters (from 15-20 m depth). However, we did not observe any clear bimodal distribution of FLR (or SWS, data not shown) signals (Fig. S4 and new Fig. S5 in supplements). The DCM (i.e. 40 m depth), where the LL ecotype is supposed to be the main ecotype, was sampled only at one occasion, during the STA11 CTD-Rosette (Fig. S2, S4 and S5). Campbell and Vaulot (1993, Fig. 4) clearly show that a bimodal distribution of FLR intensities can be observe when 2 ecotypes are present together in "similar" proportion around the DCM. By "similar", we mean a sufficient abundance of both ecotypes, which make possible to clearly identify the bimodal distribution of FLR. Blanchot and Rodier (1996) also identify such a bimodal distribution in few locations. They clearly explained that in other location, ecotypes (sub-populations) co-occurrence cannot be observed from bimodality of the FLR distribution because both ecotypes were not abundant enough to be clearly seen. In these locations both ecotypes still existed, but their concentrations were very different and thus the two peaks could not be evidenced, the larger peak overpassing the smaller one. In the revised manuscript we mention Fig. S5 and strengthen our discussion about the two Prochlorococcus ecotypes distribution over the water column.

[revised manuscript text omitted]

---

## Author Response (AR3)

The authors have made an effort in addressing the comments of the referees and the manuscript has improved. However, a number of questions should still be addressed (see below). The English language needs correction for generally minor mistakes. For example, many nouns used as adjective are in plural form, while they should be in singular form.

**Detailed comments**

Page 8, line 14. "Along most of the". Eliminate "part". Done

Page 10, line 15. Explain the meaning of the "hat" (^) above the vectors N and w. Done Line 23. "that samples". Done

Line 29 "population loss rates" instead of "population losses rates". The same problem in many other places (see comment above). Done

Page 11, line 22. "cell removal process" instead of "cells removal process", etc. I will not indicate this question again. Done

Page 11, line 8. "median" or "mean"? It was the median size ratio.

Page 13. Lines 8-9. "presented detectable variability". Presumably, there was variability, but it could not be detected. Done

Line 13. Difficult to understand. Please, rewrite. We acknowledge that the sentence was difficult to understand, we rewrote it.

Line 24. "median" or "mean"? It was the mean size, sorry for this misleading.

Line 26. "cytometry analyses". Done

Line 27. "The two counting methods did not show" instead of "Both methods counts . ." Done, thanks for this suggestion.

Page 14, Lines 4-5. "One picoeukaryote group . . . and another with high FLR" Done.

Line 12. Any relevant taxonomic identification in these pictures? We added the following statement in our manuscript: "During the campaign, the taxonomic identification based on pictures taken by the image-in-flow device was impossible due to the lack of sufficient number of phytoplanktonic cells with size above 20-30  $\mu$ m (size from which a taxonomic identification can be performed)."

Page 15, line 9. "However" instead of "although". Done

Line 27. Explain the reason for excluding the orange dots. The orange dots correspond to data acquired in type 2 warm boundary waters characterized by abnormally high FLR recorded by the AFCM compared to the TSG fluorometer (see Section 4.3.4.).

Page 17, lines 12-14. Sentence not clear, please rewrite. Done

Line 25. "distribution of cells". Done

Line 27. "modeled-produced"?? Should this be "modeled-predicted"? Yes it should. Sorry for this misleading.

Line 29. "median" or "mean"? Median.

Page 20. Line 5. "nutrient-rich". Done.

Line 12. The upper SST limit for type 1 boundary waters given in page 12, line 12 was 17°C, not 17.5°C. It was 17°C, we corrected this sentence.

Page 22, line 21. "is" instead of "are". Done.

Lines 27-29. The sentence between "The NanoHighFLO . . and lower than 5  $\mu m$ ", which refers to a subgroup of the nanoeukaryotes would be better placed after the sentence ending with

"Percopo et al., 2011". Thanks for this suggestion. Done.

Page 23, line 15. Take into account that there may seasonal variability and explain with more precision the limits of "this oligotrophic area". For example, in March 2005, at a station located at 41° 45' N, 5° 7.6'E, Estrada et al. (2014, DSR 1, 94.45-61) found Chla concentrations near 7 µg dm-3 and diatom concentrations > 100 cells cm-3. We acknowledge that when strong winter mixing episode occurs in NW Mediterranean Sea (deep convection events), strong late winter/ early spring phytoplankton, as described by Estrada et al. (2014), high Chl-a concentrations associated to high diatoms abundances can be observed. However, this kind of phenome principally occurs off the Gulf of Lion and not in the Ligurian Sea. In order to avoid any confusion, and because the main purpose of our study does not deal with seasonal microphytoplankton are never dominant in this oligotrophic area".

Page 24, line 2. What is meant by "severely light adapted"? We changed "severely light adapted" by « high-light adapted".

Line 10. "although differences of one or two orders of magnitude"? Done.

Lines 27-28. "A few measurements were not included . ." Does this refer to the orange dots? Please, indicate. Yes these measurements referred to the measurements obtained in type 2 warm boundary waters (orange dots). We rephrase this part of the discussion.

Line 29. "type 2 boundary waters". Add "boundary" to avoid confusion with optical water types. The same in page 25, line 22. Done.

Section 4.3.4. Interesting, but would be better with some streamlining. We agree that this part deserve to be streamlined. We made the requested modifications in this sense.

Page 27, lines 5-7. This statement appears to contradict that in lines 23-25 of page 21 ("Occurrence of Prochlorococcus population with significantly higher FLR (and/or SWS) values, which might be representative of the LL ecotype, was never observed in surface waters (Fig. 12 for AFCM and S4 for conventional flow cytometry)"). Please, check this. Sorry for this contradiction. This assessment was made at the beginning of our reflexion during the redaction of the paper and we omitted to delete it.

Figure 1. The labels in the axes are too small. Done.

Figures 6A and 7A. The "red color" is rather orange and the "orange color" is rather yellow, at least in my computer. On the pdf I submitted, the color distinction is quite clear. I'll have a careful look on this point before the final submission.

**Coupling physics and biogeochemistry thanks to high resolution observations of the phytoplankton community structure in the North-Western Mediterranean Sea**

Pierre Marrec1, Gérald Grégori1, Andrea M. Doglioli1, Gérald Grégori‡, Mathilde Dugenne1, Alice Della Penna1, Nagib Bhairy1, Thierry Cariou2, Sandra Hélias Nunige1, Soumaya Lahbib1, Gilles Rougier1, Thibaut Wagener1 and Melilotus Thyssen1

[revised manuscript text omitted]

the surface ( $^{5}$  m) with the MVP were compared to the data acquired from the onboard TSG. MVP temperature and salinity values were significantly correlated to the continuous underway measurements with a 1:1 relationship, R2 of 0.99 and 0.84 and root mean square error (RMSE) of residuals of 0.07°C and 0.02 for temperature and salinity, respectively.

A total of 8 fixed stations were performed (Fig. 2) and used to collect biogeochemical information and to validate the deployment of the MVP. For each station, a CTD-rosette cast down to 300 m recorded temperature, salinity and fluorescence profiles. At Station 11, the water column properties down to 1000 m wereas investigated with this CTD-rosette instrument. The CTD-rosette was equipped with a 12 Niskin bottle (12 dm3) SBE32 Carousel water sampler and carried a CTD SBE911+ for temperature and salinity, a Chelsea Aquatracka III fluorimeter and a QCP-2350 (cosinus collector) for PAR measurements. Samples for nutrients and phytoplankton groups using bench-top flow cytometry (Sect. 2.4.) were collected from the surface to 1000\_-m depth.

For stations 5 to 11 (Fig. 2), an innovative system of high-resolution seawater sampling down to 35 m (PASTIS HVR – Pumping Advanced System To Investigate Seawater with High Vertical Resolution) was deployed. Seawater samples were collected using a Teflon pump (AstiPureTM II High Purity Bellows Pumps – Flow rate =  $30 \text{ dm}^3$ .min-1) connected to a polyethylene (PE) tube fixed to the frame at the level of the pressure sensor of a Seabird SBE19+ CTD and a WetLab WETstar WS3S fluorimeter. The depth of the sampling was defined as the mean depth recorded by the pressure sensor with a vertical resolution of 0.1 to 1 m (depending on the sea state). The SBE19+ CTD offered precisions for temperature and computed salinity of 0.005°C and 0.002, respectively. The PASTIS HVR was used to collect samples every 2-3 m for bench-top flow cytometry analyses (Sect. 2.4.). Complementary nutrient analyses were made at a lower vertical resolution (10 m). Nitrite and phosphate concentration profiles never overpassed the limits of quantification of the analyzers (data not shown). Random 27 seawater samples were collected and filtered to measure Chl-a concentration (Sect. 2.4.) and to convert fluorescence signal into Chl-a values. A significant correlation between fluorescence and Chl-a was obtained with an R2 of 0.52 (pvalue<0.05). A cross-calibration in terms of fluorescence was performed between fluorometers of the CTD-Rosette and the CTD used for PASTIS\_HVR to harmonize Chl-a values (fluorescenceCTD PASTIS HVR = fluorescenceCTD\_rosette x 3.31, n=60, R2=0.85).

**2.7 Surface specific growth rates and primary production estimates**

Phytoplankton growth rates were estimated by measuring independently with AFCM the net abundances combined with a size-structured population model described in Sosik et al. (2003) and adapted by Dugenne et al. (2014) and Dugenne (2017). Observed diel variations of single cell biovolumes within a specific cluster, retrieved from the power law relationship between cell size and FWS, were used as inputs for this size-structured population model. The absolute number of cells ( $\vec{N}$ ) and proportions of cells ( $\vec{W}$ ) were counted during 24 hours to follow the transitions of cells in each size class ( $\nu$ ).

$$\vec{N} = \begin{pmatrix} N_{1|\nu=\nu_{1}} \\ \vdots \\ N_{i|\nu=\nu_{i}} \\ \vdots \\ N_{m|\nu=\nu_{m}} \end{pmatrix}, \vec{w} = \frac{\vec{N}}{\sum_{i=1}^{m} N_{i|\nu=\nu_{i}}}$$
(1)

with  $v_{1,\dots,i,\dots,m}$  denoting the size classes.

We identified the set of parameters that could optimally reproduce the diel variation of population size distribution using only cell cycle transitions by inverse modelling. In the model, temporal transitions of cells proportions in size classes are assumed to result from either cellular growth, supported by photosynthetic carbon assimilation, or asexual division. The increase of cell size occurring during the interphase is dependent of the proportions of cells that will grow between t and t + dt, noted y(t). This probability is expressed as an asymptotic function of incident irradiance (Eq. (2)).

$$\gamma(t) = \gamma_{max} \cdot \left[1 - exp\left(-\frac{Irradiance}{Irradiance^*}\right)\right]$$

[revised manuscript text omitted]

The apparent increase of carbon biomass, defined as the Net Primary Production NPPcell (Eq. (9), mg C.m-3.d-1), was calculated using a constant cell to carbon conversion factor QC, calc (Table 2).

$$NPP_{cell} = Q_c.\,\delta(t).\,\vec{N}(T_0) = Q_c.\,[\exp(\mu(t)) - 1].\,\vec{N}(T_0)$$
(9)

\_\_\_\_\_The biovolume to carbon  $av_i^b$  relationship (Table 2), was used to calculate the Net Primary Production NPPsize (Eq. (10)) as the differential of carbon distributions, as the scalar product of vectors  $av_i^b$  and N over time:

$$NPP_{size} = \sum_{t \in \mathbb{R}^*} \Delta(\langle C_{size}, \vec{N}(t+dt) \rangle, \langle C_{size}, \vec{N}(t) \rangle)$$

[revised manuscript text omitted]

Table 2. Mean and standard deviation of forward scatter (FWS), equivalent spherical diameter (ESD) and biovolume of *Prochlorococcus*, *Syneenchococcus*, PicoEukaryotes (PicoE) and NanoEukaryotes (NanoE) during the OSCAHR campaign. ESD were computed according to the power law relationship (log(Size)=0.309\*log(FWS)-1.853, n = 17, r2 = 0.94) obtained with silica beads of known diameter. Biovolumes were calculated considering that the cells were spherical. Biovolumes were converted into mean carbon cellular quota ( $Q_{C, calc}$ ) according to the  $Q_{C, calc}$ =a.Biovolumeb relationship using conversion factors a and b reported by (1) Menden-Deuer and Lessard (2000). Carbon cellular quota ( $Q_{C, lit}$ , lit for literature) from (2) Campbell et al. (1994), (3) Shalapyonok et al. (2001) and (4) Reifel et al. (2014) were reported for comparison.

|                                                  | Prochlorococcus           | Synechococcus                      | PicoEukaryotes            | NanoEukaryotes              |
|--------------------------------------------------|---------------------------|------------------------------------|---------------------------|-----------------------------|
| FWS (a.u. cell -1 )                   | 48 ± 21                   | 357 ± 335                          | $1.0\ 10^4\pm 0.6\ 10^4$  | $4.0\ 10^4\pm1.7\ 10^4$     |
| ESD (μm)                                         | $0.5 \pm 0.1$             | $0.9 \pm 0.2$                      | $2.6 \pm 0.5$             | $4.1 \pm 0.5$               |
| Biovolume (µm 3 .cell -1 ) | $0.07 \pm 0.03$           | $0.46 \pm 0.38$                    | 10.5 ± 5.5                | 37.0 ± 14.7                 |
| Conversion coefficients (a, b)                   | (0.26, 0.86) 1 | (0.26, 0.86) 1          | (0.26, 0.86) 1 | (0.433, 0.863) 1 |
| $Q_{C, calc.}$ (fg C cell -1 )        | 25                        | 109                                | 1880                      | 9000                        |
| $Q_{C, lit.}$ (fg C cell -1 )         | 53 2           | 100 3 -250 2 | 2108 2         | 9160 4           |

Table 3. *Prochlorococcus* and *Synechococcus* daily growth rate estimate ( $\mu_{ratio}$ ) computed as the median size ratio  $\mu_{ratio} = ln(\overline{v}_{max}/\overline{v}_{min})$ , intrinsic growth rate ( $\mu_{size}$ ) and loss rate (l) obtained from Eq. 7. NPPcell and NPPsize biomass production values obtained from Eq. 8 and 9, respectively.

|                                                         | Prochlorococcus | Synechococcus |
|---------------------------------------------------------|-----------------|---------------|
| $\mu_{ m ratio}$ (d -1 )                     | 0.28            | 0.49          |
| $\mu_{ m size}$ (d -1 )                      | 0.21            | 0.72          |
| / (d -1 )                                    | 0.30            | 0.68          |
| $NPP_{cell}$ (mg C. m -3 . d -1 ) | 0.11            | 2.68          |
| $NPP_{size}$ (mg C. m -3 . d -1 ) | 0.13            | 2.80          |